# CLOSED-FORM LAST LAYER OPTIMIZATION

## ABSTRACT

Neural networks are typically optimized with variants of stochastic gradient descent. Under a squared loss, however, the optimal solution to the linear last layer weights is known in closed-form. We propose to leverage this during optimization, treating the last layer as a function of the backbone parameters, and optimizing solely for these parameters. We show this is equivalent to alternating between gradient descent steps on the backbone and closed-form updates on the last layer. We adapt the method for the setting of stochastic gradient descent, by trading off the loss on the current batch against the accumulated information from previous batches. Further, we prove that, in the neural tangent kernel regime, convergence of this method to an optimal solution is guaranteed. Finally, we demonstrate the effectiveness of our approach compared with standard SGD on a squared loss in several supervised tasks – both regression and classification – including Fourier Neural Operators and Instrumental Variable Regression.

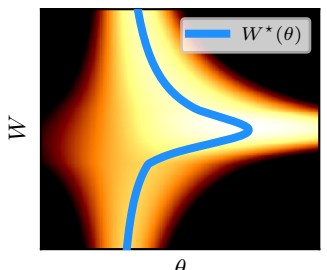

Figure 1: The squared loss landscape of a two-parameter neural network

$$f(x) = W \operatorname{ReLU}(\theta x)$$

with three random training data points. Dark / light regions correspond to values of high / low loss respectively. We plot in blue the optimal last layer parameter $W^\star(\theta)$ as a function of the backbone parameter $\theta$. We propose to optimize along the blue curve, rather than in two-dimensional space.

## 1 INTRODUCTION

Training deep neural networks is almost always done with variants of stochastic gradient descent (SGD). Despite their empirical success, these iterative methods treat every layer of the network in the same way. However, the linear last layer often admits a much simpler – and in the case of squared loss, closed-form – optimal solution. This mismatch suggests an opportunity: if the optimal last layer weights can be computed directly given the current features produced by the backbone, we can regard the last layer as an implicit function of the backbone parameters. This could simplify the optimization problem by constraining the last layer to be optimal throughout (see Fig. 1).

In SGD, gradients at each step are computed in minibatches. Because of computational constraints, the closed-form solution of the last layer should also use minibatches. This risks overfitting the last layer to each batch at each optimization step. To correct for this issue, there is the need to account for previous last layer solutions.

In this paper, we develop a training procedure that can perform optimization with a closed-form optimal last layer through SGD on the backbone parameters. Our contributions are as follows:

1. We propose leveraging the closed-form last layer solution for squared loss, and optimizing the backbone parameters while treating the last layer as a deterministic function of those parameters. We show that this requires no backpropagation through the closed-form solution (Section 3).

2. We adapt the approach to stochastic mini-batch training by regularizing for previous last layer solutions, producing a practical algorithm that integrates cleanly with standard training pipelines and that admits an approximate Kalman filter interpretation (Section 4).

3. We provide a theoretical analysis in the infinite width neural tangent kernel (NTK) limit, proving convergence of the method to an optimal solution, in the deterministic and continuous time case (Section 5).

4. We validate the approach empirically, demonstrating improvements compared to standard training under squared losses, including applications in deep feature instrumental variable regression and Fourier neural operators (Section 6).

## 1.1 RELATED WORK

We outline several strands of related work.

**Two-timescale regime.** Optimizing under a closed-form last layer can be seen as performing bilevel optimization, where an optimization problem is nested into another (Zhang et al., 2024; Petrulionyte et al., 2024). In recent years, this last layer bilevel optimization approach has been considered in several works as a simplifying assumption for demonstrating convergence of gradient descent in neural networks. This framework was coined the two-timescale regime (Marion & Berthier, 2023; Berthier et al., 2024; Bietti et al., 2025; Barboni et al., 2025).

Marion & Berthier (2023) noted that, by the envelope theorem, optimizing with an optimal last layer as a function of the backbone parameters is equivalent to optimizing only the backbone parameters while keeping the last layer optimal. In the present work, we bring this theoretical argument to a practical method that can accelerate optimization, by observing that the envelope result can be leveraged computationally in backpropagation. Indeed, unlike these works, our aim is to propose novel methodology, and demonstrate its practicality in a number of scenarios. From the theoretical side, our work is the first to analyze the critical points of the resulting loss in function space, and the convergence to a global minimum in the NTK regime. Other works have investigated the mean-field regime instead (Wang et al., 2024; Takakura & Suzuki, 2024).

In an experiment, Barboni et al. (2025) propose to update the last layer by an exponential moving average of closed-form solutions to account for the stochasticity in SGD. However, this approach decouples the last layer from the backbone, as they no longer optimize the same loss, which leads to instabilities. In contrast, our approach for stochasticity allows the last layer and the backbone to continue optimizing for the same loss.

**Layer-wise learning.** Singh et al. (2015); You et al. (2017) propose to tune the learning rates of SGD layer-wise. Without the regularization to previous last layer solutions, our method is analogous to putting a large learning rate on the last layer. You et al. (2017) further show that, while shallower layers tend to have smaller gradients, this is not a reason for such layers needing larger learning rates, as the layer weights also appear to be smaller themselves. Indeed, Chen et al. (2022a) analyze the convergence speeds of different layers during optimization, and show that, despite smaller gradients, shallower layers learn faster than deeper layers. Our method thus remediates this layer convergence bias for the last layer.

**Bayesian last layers.** These works leverage closed-form or quasi-closed-form solutions for last layer Bayesian posteriors to train them with variational inference (Harrison et al., 2023; Brunzema et al., 2024; Harrison et al., 2025). In contrast, the present work does not construct a Bayesian posterior, and instead leverages closed-form solutions for the last layer to accelerate optimization.

**Feature learning in instrumental variables regression.** Training with a closed-form last layer has found applications in instrumental variables regression (Xu et al., 2020; You et al., 2017). In these applications, the closed-form last layer solution is required to solve the bilevel optimization problem. However, these works backpropagated through the closed-form solution, making them computationally expensive. Moreover, they did not employ a regularized solution, and thus required large batch sizes for the networks to train. We address both limitations in the present work, and demonstrate superior performance of our method in these applications.

## 2 BACKGROUND

**Non-linear multi-dimensional regression.** We consider the regression problem with a squared loss in which we aim to predict $y \in \mathbb{R}^o$ from the input $x \in \mathcal{X}$ (typically $\mathcal{X} \subset \mathbb{R}^m$), where $\mathcal{X}$ is

some input space. We employ a model $f(x; W, \theta) = W\phi_\theta(x)$, where $\phi_\theta \colon \mathcal{X} \to \mathbb{R}^d$ is the neural network feature map, which we call a *backbone*, parametrised by $\theta \in \Theta$, $\Theta \subset \mathbb{R}^N$ is the feature parameter space, and $W \in \mathbb{R}^{o \times d}$ is the weight matrix for the last linear layer. Given $n$ observations $\{(x_i, y_i)\}_{i=1}^n$, we want the find the parameters $(W, \theta)$ that minimize the regularized squared loss

$$\mathcal{L}(W, \theta) = \sum_{i=1}^n \|y_i - W\phi_\theta(x_i)\|_2^2 + \beta\|W\|_F^2 \tag{1}$$

for some hyperparameter $\beta > 0$, where $\|\cdot\|_F$ is the Frobenius norm.

For a fixed $\theta$, the objective Eq. (1) is a ridge regression problem which enjoys the closed-form solution

$$W^\star(\theta) := \underset{W \in \mathbb{R}^{o \times d}}{\arg\min} \mathcal{L}(W, \theta) = Y\phi_\theta(X)^\top \left(\phi_\theta(X)\phi_\theta(X)^\top + \beta I\right)^{-1} \tag{2}$$

where $X = (x_1, \ldots, x_n)$, $Y = (y_1, \ldots, y_n) \in \mathbb{R}^{o \times n}$ and $\phi_\theta(X) \in \mathbb{R}^{d \times n}$ is a matrix where we apply $\phi_\theta$ to each $x_i$.

**Gradient descent.** An approach to optimize Eq. (1)) for $W$ and $\theta$ is to use gradient descent, an iterative optimization algorithm which starts from $(W_0, \theta_0)$ and at each iteration $t$ solves the following linearized optimization problem

$$W_{t+1}, \theta_{t+1} = \underset{W, \theta}{\arg\min} \nabla\mathcal{L}(W_t, \theta_t)^\top [W, \theta] + \frac{1}{2\alpha}\|W - W_t\|_F^2 + \frac{1}{2\eta}\|\theta - \theta_t\|_F^2, \tag{3}$$

where $(\alpha, \eta)$ are learning rates and $[W, \theta]$ denotes the concatenation operation. The solution to Eq. (3) can be expressed with the familiar gradient descent updates

$$W_{t+1} = W_t - \eta\nabla_W\mathcal{L}(W_t, \theta_t), \quad \theta_{t+1} = \theta_t - \alpha\nabla_\theta\mathcal{L}(W_t, \theta_t). \tag{4}$$

## 3 Optimizing with a Closed-Form Last Layer

During optimization, we would like to leverage the fact that, for each $\theta$, the optimal last layer $W^\star(\theta)$ is available in closed-form (Eq. (2)). The idea is that there is no need to update $W_t$ through gradient steps as in Eq. (4), as we may treat it directly as a function of $\theta$ through Eq. (2). This leads to the loss

$$\mathcal{L}^\star(\theta) := \mathcal{L}(W^\star(\theta), \theta) = \sum_{i=1}^n \|y_i - W^\star(\theta)\phi_\theta(x_i)\|_2^2 + \beta\|W^\star(\theta)\|_F^2 \tag{5}$$

We now propose to optimize this loss instead of Eq. (1). Computationally, this involves alternating between solving linear regressions to obtain $W^\star(\theta)$ and gradient steps on $\theta$ through $\mathcal{L}$ (backpropagating through the closed-form solution to the regression). Explicitly, we start at some $\theta_0$ and iterate

$$W_{t+1} = W^\star(\theta_t), \quad \theta_{t+1} = \theta_t - \alpha\nabla_\theta\mathcal{L}^\star(\theta_t). \tag{6}$$

Note that $\nabla_\theta\mathcal{L}^\star(\theta_t)$ involves backpropagating through $W^\star(\theta)$ given by Eq. (2), and hence through an inverse. This is computationally demanding. Fortunately, this operation is not needed, as the following theorem demonstrates (see also the *envelope theorem* and Marion & Berthier (2023, Remark 1)):

**Theorem 1.** *For fixed $\theta$, letting $W^\star := W^\star(\theta)$ with Eq. (2), we have*

$$\nabla_\theta\mathcal{L}^\star(\theta) = \nabla_\theta\mathcal{L}(W^\star, \theta) \tag{7}$$

*Proof.* By the chain rule,

$$\nabla_\theta\mathcal{L}^\star(\theta) = \underbrace{\nabla_W\mathcal{L}(W^\star, \theta)}_{=0} \mathrm{D}W^\star(\theta)^\top + \nabla_\theta\mathcal{L}(W^\star, \theta)\underbrace{\mathrm{D}\,\mathrm{id}(\theta)^\top}_{=I} = \nabla_\theta\mathcal{L}(W^\star, \theta) \tag{8}$$

where $\nabla_W\mathcal{L}(W^\star, \theta) = 0$ follows from the fact that $W^\star = \arg\min_W \mathcal{L}(W, \theta)$ and $\mathrm{D}$ denotes the differential operator. $\qquad\square$

Compared to $\nabla_\theta \mathcal{L}^\star(\theta)$ which requires a complicated backpropagation, $\nabla_\theta \mathcal{L}(W^\star, \theta)$ requires just a usual backpropagation through $\phi_\theta$, keeping the last layer $W^\star$ fixed. Theorem 1 thus shows that Eq. (6) is equivalent to

$$W_{t+1} = W^\star(\theta_{t+1}), \quad \theta_{t+1} = \theta_t - \alpha \nabla_\theta \mathcal{L}(W_{t+1}, \theta_t), \tag{9}$$

i.e. it suffices to replace the gradient step on $W$ in Eq. (4) by a closed-form update of the form Eq. (2).

## 4 THE STOCHASTIC SETTING

In practice, neural networks are not trained with gradient descent, but with *stochastic* gradient descent, or variants thereof. At time $t$ we observe a batch of data $\mathcal{B}_t \subset \{(x_i, y_i)\}_{i=1}^n$. Then the squared loss on the batch is given by

$$\mathcal{L}_{\mathcal{B}_t}(W, \theta) := \sum_{(x_i, y_i) \in \mathcal{B}_t} \|y_i - W\phi_\theta(x_i)\|_2^2 + \beta \|W\|_F^2. \tag{10}$$

Similarly we write

$$W_{\mathcal{B}_t}^\star(\theta) := \underset{W \in \mathbb{R}^{o \times d}}{\arg\min} \mathcal{L}_{\mathcal{B}_t}(W, \theta). \tag{11}$$

Naively, in the stochastic setting, we might use Eq. (9) but with $\mathcal{L}$ replaced by $\mathcal{L}_{\mathcal{B}_t}$ and $W^\star$ replaced $W_{\mathcal{B}_t}^\star$. While such an approach will be valid for large batch sizes, it will be ineffective for small batches as the last layer $W_{\mathcal{B}_t}^\star(\theta)$ will overfit to each batch $\mathcal{B}_t$ at each $t$. The last layer estimates $W_{t+1}$ (see Eq. (9)) will then vary drastically at each iteration. As a consequence, the features $\phi_{\theta_t}(X)$ might not be able to adapt to an unstable last layer. The smaller the batch size relative to the complete dataset, the more severe the issue (see the experiments in Section 6).

Instead, we propose to optimize a different loss. Motivated by how gradient descent regularizes to the previous estimates of the parameters (see Eq. (3)) we propose to regularize the objective function against the distance from $W$ to the previous estimate $W_t$, yielding the *proximal loss*

$$\mathcal{L}_{\mathcal{B}_t, W_t}^{\text{prox}}(W, \theta) := \sum_{(x_i, y_i) \in \mathcal{B}_t} \|y_i - W\phi_\theta(x_i)\|_2^2 + \lambda \|W - W_t\|_F^2 \tag{12}$$

where $\|\cdot\|_F$ is the Frobenius norm and $\lambda > 0$ is some hyperparameter. This ensures that closed-form solutions to Eq. (12) are close to the previous estimate $W_t$.

As before, we define

$$W_{\mathcal{B}_t, W_t}^\star(\theta) = \underset{W \in \mathbb{R}^{o \times d}}{\arg\min} \mathcal{L}_{\mathcal{B}_t, W_t}^{\text{prox}}(W, \theta) = \left(Y\phi_\theta(X)^\top + \lambda W_t\right)\left(\phi_\theta(X)\phi_\theta(X)^\top + \lambda I\right)^{-1} \tag{13}$$

and

$$\mathcal{L}_{\mathcal{B}_t, W_t}^{\text{prox}\,\star}(\theta) := \mathcal{L}_{\mathcal{B}_t, W_t}^{\text{prox}}(W_{\mathcal{B}_t, W_t}^\star(\theta), \theta). \tag{14}$$

Thus we propose to start at some $(W_0, \theta_0)$ and iterate

$$W_{t+1} = W_{\mathcal{B}_t, W_t}^\star(\theta_t), \quad \theta_{t+1} = \theta_t - \alpha \nabla_\theta \mathcal{L}_{\mathcal{B}_t, W_t}^\star(\theta_t). \tag{15}$$

This approach addresses the stochasticity issues, while ensuring that $W$ and $\theta$ optimize for the same loss, namely $\mathcal{L}^{\text{prox}}$. Moreover, we can obtain a result analogous to Theorem 1 for the proximal loss:

**Theorem 2.** *For fixed $\theta$, letting $W_{\mathcal{B}_t, W_t}^\star := W_{\mathcal{B}_t, W_t}^\star(\theta)$, we have*

$$\nabla_\theta \mathcal{L}_{\mathcal{B}_t, W_t}^{\text{prox}\,\star}(\theta) = \nabla_\theta \mathcal{L}_{\mathcal{B}_t}(W_{\mathcal{B}_t, W_t}^\star, \theta). \tag{16}$$

*Proof.* Arguing as for Theorem 1, we have $\nabla_\theta \mathcal{L}_{\mathcal{B}_t, W_t}^{\text{prox}\,\star}(\theta) = \nabla_\theta \mathcal{L}_{\mathcal{B}_t, W_t}^{\text{prox}}(W_{\mathcal{B}_t, W_t}^\star, \theta)$. Now since the regulariser $\|W_{\mathcal{B}_t}^\star - W_t\|$ does not depend on $\theta$, we have $\nabla_\theta \mathcal{L}_{\mathcal{B}_t, W_t}^{\text{prox}}(W_{\mathcal{B}_t, W_t}^\star, \theta) = \nabla_\theta \mathcal{L}_{\mathcal{B}_t}(W_{\mathcal{B}_t, W_t}^\star, \theta)$. □

Like Theorem 1, Theorem 2 allows us to replace the demanding backpropagation procedure to compute $\nabla_\theta \mathcal{L}_{\mathcal{B}_t, W_t}^{\text{prox}\,\star}(\theta)$ by a classical backpropagation step for $\nabla_\theta \mathcal{L}_{\mathcal{B}_t}(W_{\mathcal{B}_t}^\star, \theta)$. So Eq. (15) is equivalent to

$$W_{t+1} = W_{\mathcal{B}_t, W_t}^\star(\theta_t), \quad \theta_{t+1} = \theta_t - \alpha \nabla_\theta \mathcal{L}_{\mathcal{B}_t}(W_{t+1}, \theta_t). \tag{17}$$

In Appendix A we give another interpretation for Eq. (17) as doing approximate Kalman filtering on the last layer throughout SGD on the backbone parameters.

## 4.1 NUMERICAL CONSIDERATIONS

**Use of a bias term.** We could add an additional bias dimension to feature vector $\phi_\theta$ and a learnable bias $b$ to the last layer $W$, which lead to extended vectors $\tilde{\phi}_\theta = [\phi_\theta, 1]$ and $\tilde{W} = [W, b]$.

**Last layer initialization.** First, we consider *zeros*, i.e. $W_0^{ij} = 0, \forall i, j$, which worked the best in practice. We then consider classical weight initializations – *LeCun normal* $W_0^{ij} \sim \mathcal{N}\left(0, \frac{1}{d}\right)$, *Xavier normal* $W_0^{ij} \sim \mathcal{N}\left(0, \frac{2}{d+o}\right)$ and *He normal* $W_0^{ij} \sim \mathcal{N}\left(0, \frac{2}{d}\right)$. Bias is always initialized as $b_0^i = 0$.

**Full algorithm** In practice, when we use Eq. (17), the backbone parameters $\theta$ will always be more "up-to-date" than the last layer parameters, because the backbone is updated after the last layer. In this case, at time $t$, the performance evaluated with $(W_t, \theta_t)$ might be sub-optimal. We propose two different approaches in order to correct for this. In one approach, we structure the algorithm so that the last layer is always up-to-date by updating the backbone parameters on the current batch and the last layer parameters on the future batch. We describe this approach in Algorithm 2 in Appendix B. Empirically, we found that an alternative approach worked better. In this variant, we simply update the backbone first, and then the last layer on the current batch. It is simpler than the previous method and can be easily plugged in the existing optimizers code. We describe this in Algorithm 1.

---

**Algorithm 1** Simple proximal closed-form SGD

1: **Given:** Batch size $B$, proximal coefficient $\lambda > 0$, neural network $\phi_\theta$ with initial parameters $\theta_0$, learning rate $\alpha > 0$, initial last layer parameters $W_0$.
2: $t \leftarrow 0$
3: **while** $\theta_t$ has not converged **do**
4:      $t \leftarrow t + 1$
5:      **Update backbone on the current batch $\mathcal{B}_t$**
6:      $\theta_t \leftarrow \theta_{t-1} - \alpha \nabla_\theta \mathcal{L}_{\mathcal{B}_t}(W_{t-1}, \theta_{t-1})$
7:      **Update last layer on the current batch $\mathcal{B}_t$**
8:      $W_t \leftarrow W^\star_{\mathcal{B}_t, W_{t-1}}(\theta_t)$
9: **Output:** Optimized $(W^\star, \theta^\star)$

---

Note that by swapping the order in which the backbone and last layer are updated, Algorithm 1, unlike Algorithm 2, slightly departs from Eq. (17) and Theorem 2.

## 4.2 APPLICATION TO CLASSIFICATION

In classification, we treat the output $y_i$ as one-hot vectors, i.e., $y_i \in \{0, 1\}^C$ such that $\sum_{c=1}^{C} y_i^c = 1$, where $C = o$ is the number of classes. We then optimize a squared loss (see for instance Hui & Belkin (2021) for the use of squared loss in classification). We use the strategy Eq. (17) to optimize $W$ and $\theta$. However, optimizing in this way does not guarantee that the model $f(x; W, \theta) = W\phi_\theta(x)$ outputs probability vectors, i.e. $\sum_{c=1}^{C} W^c \phi_\theta(x) \neq 1$, where $W^c$ is the $c^{\text{th}}$ row of $W$. Therefore, for prediction, we simply take the arg max over output vectors, i.e. $c_{\text{out}}(x) = \arg\max_c W^c \phi_\theta(x)$. While this strategy is a simple heuristic, we found that using it together with Eq. (17) led to reasonable performance.

## 5 THEORETICAL ANALYSIS OF THE LOSS

In the section we uncover theoretical insights for the loss $\mathcal{L}^\star(\theta)$ from Eq. (5). For a tractable analysis, we will start by considering this loss as a function of the backbone, instead of the parameters $\theta$.

Let $\mathcal{F}$ be the space of functions $\phi : \mathcal{X} \to \mathbb{R}^d$. Then we define the loss Eq. (1) but taking a backbone $\phi$ as second argument:

$$\mathcal{L}_\mathcal{F}(W, \phi) = \|y_i - W\phi(x_i)\|_2^2 + \beta\|W\|_F^2 \tag{18}$$

where $W \in \mathbb{R}^{o \times d}$, $\phi \in \mathcal{F}$ and $\beta > 0$. Then, as before we define

$$W^\star_\mathcal{F}(\phi) := \underset{W \in \mathbb{R}^{o \times d}}{\arg\min} \mathcal{L}_\mathcal{F}(W, \phi) = Y\phi(X)^\top \left(\phi(X)\phi(X)^\top + \beta I\right)^{-1} \tag{19}$$

and

$$\mathcal{L}_{\mathcal{F}}^{\star}(\phi) := \mathcal{L}_{\mathcal{F}}^{\star}(W_{\mathcal{F}}^{\star}(\phi), \phi) = \sum_{i=1}^{n} \|y_i - W_{\mathcal{F}}^{\star}(\phi)\phi(x_i)\|_2^2 + \beta\|W_{\mathcal{F}}^{\star}(\phi)\|_F^2. \tag{20}$$

$\mathcal{L}_{\mathcal{F}}^{\star}$ possesses unexpected characteristics. One set of critical points of $\mathcal{L}_{\mathcal{F}}^{\star}$ are the minimizers $\phi^{\star}$, which perfectly balance between fitting the data $W_{\mathcal{F}}^{\star}(\phi^{\star})\phi^{\star}(x_i) \approx y_i$ and the regularizer controlled by $\beta$. However, these are not all critical points.

**Theorem 3.** *If $Y \neq 0$ then $\mathcal{L}_{\mathcal{F}}^{\star}$ is not convex, and it admits critical points $\phi^{\star}$ that are not global minimizers.*

The proof for this theorem can be found in Appendix C.

In contrast, the usual squared (or ridge) loss $\sum_{i=1}^{n} \|y_i - f(x_i)\|_2^2$, where $f \colon \mathcal{X} \to \mathbb{R}^o$ which does not use a closed-form solution on the last layer $W_{\mathcal{F}}^{\star}(\phi)$, is convex in $f$ . The critical points of this loss function are exactly the functions $f^{\star}$ such that $f^{\star}(x_i) = y_i$ for all $i$ (or $f^{\star}(x_i) \approx y_i$ in the absence of a ridge regularizer).

The non-trivial critical points of $\mathcal{L}_{\mathcal{F}}^{\star}$ occur because, when the features $\phi(X)_j$ are orthogonal to all the outputs $Y_{k}$, there is no gradient information for the features. For example $\phi^{\star} = 0$ is always a critical point of $\mathcal{L}_{\mathcal{F}}^{\star}$.

Is this an issue when $\phi$ is a neural network? We next analyze the loss in the neural tangent kernel (NTK) infinite width neural network regime Jacot et al. (2018). In this regime, the initial function neural network function $\phi$ can be shown to be a Gaussian process with respect to the random initialization, controlled by the neural Gaussian process kernel (NGPK). The training dynamics of $\phi$ are given by kernel gradient descent in function space $\mathcal{F}$ with respect to the NTK. If the NTK is positive definite, we know that $\phi$ will converge to a critical point of $\mathcal{L}_{\mathcal{F}}^{\star}$ in $\mathcal{F}$. The following result shows that if we make the slightly stronger assumption that the NGPK is positive definite (see for example Gao et al. (2023, Theorem 4.5)), then $\phi$ will converge to a global minimizer.

**Theorem 4.** *In the NTK regime with positive definite NGPK, assuming $\min(d, n') \geq \operatorname{rank} Y$ where $n'$ is the number of distinct $x_i$, $\mathcal{L}_{\mathcal{F}}^{\star}$ converges almost surely to a global minimizer.*

The assumption $\min(d, n') \geq \operatorname{rank} Y$ simplifies the proof, as it ensures that the outputs $Y$ are expressible through the features $\phi(X)$, which have maximal rank $\min(d, n')$. See Appendix C.1 for a formal description of the NTK regime and a proof of this theorem.

## 6 EXPERIMENTS

**Studied methods.** First, we consider our proximal closed-form approach Eq. (17), which we call "$\ell_2$ *c.f. proximal ($\lambda$)*". Then, we also study the closed-form ridge regression approach (Eq. (9)), "$\ell_2$ *c.f. ridge ($\beta$)*". As baseline, we report performance of "$\ell_2$ *loss*", which optimizes the $\ell_2$ loss Eq. (10) with SGD. Finally, whenever it is suitable, we report performance of "*Cross Entropy*" which uses SGD. For the experiments in the main paper, we report performance of Algorithm 1 since we found it worked better in practice. The results for Algorithm 2 are provided in Appendix F. For most of the experiments, we study performance with different batch sizes. We expect "$\ell_2$ *c.f. proximal ($\lambda$)*" to be effective across batch sizes while "$\ell_2$ *c.f. ridge ($\beta$)*" to only work well with large batch sizes.

**Hyperparameters.** Unless specified otherwise, for closed-form methods we use an additional bias term and the *zeros* initialization. We always sweep over method-specific hyperparameters ($\lambda$ or $\beta$) as well as the learning rate $\alpha$. Every experiment is run with 3 random seeds (unless specified otherwise), and as a selection criterion, we compute the average performance across seeds (either loss or accuracy) at the end of the training. We train the models on *training set* and we use a separate *validation* set for selecting hyperparameters. The performance is reported on a hold-out *test* set and the dashed regions denote the $95\%$ confidence interval.

**Regression.** We consider the Fourier Neural Operator (FNO) setting described in (Li et al., 2021) applied to 1d Burgers equation. We refer the reader to the github repository (Koehler, 2024) which we used for our experiments. We consider the equation

$$\frac{\partial u}{\partial t} + \frac{1}{2}\frac{\partial u^2}{\partial x} = \nu\frac{\partial^2 u}{\partial x^2} \tag{21}$$

on the domain $\Omega = (0, 2\pi)$ where the solution is periodic ($u(t, x = 0) = u(t, x = 2\pi)$) and $\nu = 0.1$. Our dataset consists of 2048 initial conditions $u(t = 0, x)$ on a $N = 8192$ resolution grid together with their solution at time one $u(t = 1, x)$. The data is split into training (1448 points), validation (200 points) and test (400 points) sets. We train for 100 epochs on the 32-fold downsampled resolution grid (256 DoFs instead of 8192). The last layer $W \in \mathbb{R}^{H \times 1}$ is shared for every resolution dimension. We report the mean squared error (MSE) on the whole $N = 8192$ grid.

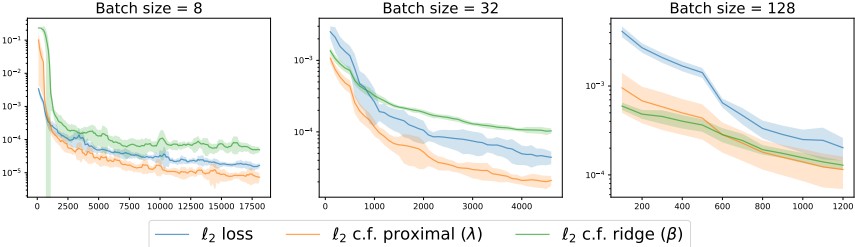

Figure 2: **Regression results**. X-axis is the number of iterations, Y-axis is a test set mean squared error (MSE), columns represents different batch sizes. Different colors indicate different methods. We use a rolling average with window size 5 to smooth the curves.

The results are provided in Figure 2. Our approach "$\ell_2$ *c.f. proximal* ($\lambda$)" outperforms "$\ell_2$ *loss*" across batch sizes. The method "$\ell_2$ *c.f. ridge* ($\beta$)" is worse than "$\ell_2$ *c.f. proximal* ($\lambda$)" for small batch sizes, but matches performance for large batch sizes. This is expected because in the large batch size regime, the objective (10) is close to the full dataset setting (1,) where the proximal term is not necessary.

**Deep Feature Instrumental Variable (DFIV) regression.** We conduct experiments in a causal two-stage regression setting. We refer the reader to Appendix D for more information. The experimental details are provided in Appendix E. In this two-stage regime, we adapt DFIV (Xu et al., 2020) to a minibatch setting and we run our proximal variant which we call "*DFIV Proximal*". For evaluation, we use two strategies. The first follows (Xu et al., 2020) and re-estimates first-stage and second-stage last layers with ridge regression (we use 0.01 coefficient for this) on the whole training set. The second strategy uses the current estimates of the last layers. The results are given in Fig. 3. We observe that for small batches, similar to the previous section, our method outperforms DFIV and achieves a similar performance in a large batch regime. An interesting feature of our method is that the performance of the second strategy is very close to the first strategy, which removes the need to re-estimate the last layers on the whole training set.

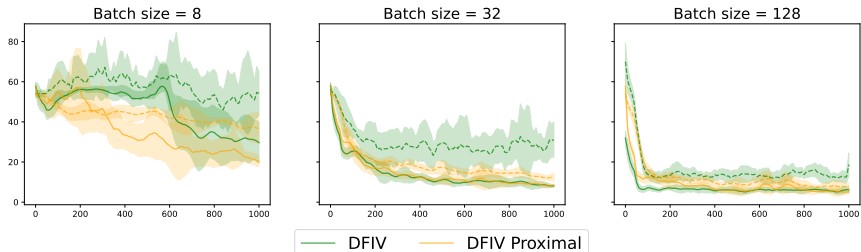

Figure 3: **DFIV results**. X-axis is the number of iterations, Y-axis is a test set MSE. Each column corresponds to a different batch size. Different colors indicate different methods. Solid lines use the last layer re-estimated on the entire training set, while dashed lines use current last layer estimates. We use a rolling average with window size 5 to smooth the curves.

**Application to classification.** We perform experiments on the CIFAR-10 and CIFAR-100 datasets (Krizhevsky & Hinton, 2009), across batch sizes $B = [32, 128, 1024, 4096]$, where we use ResNet-18 (He et al., 2016) as a backbone $\phi_\theta$. Please refer to Appendix E for more details.

The results are presented for CIFAR-10 in Figure 4 and for CIFAR-100 in Figure 5. Our method "$\ell_2$ *c.f. proximal* ($\lambda$)" performs better than "$\ell_2$ *loss*" approach in both cases, as in the regression setting.

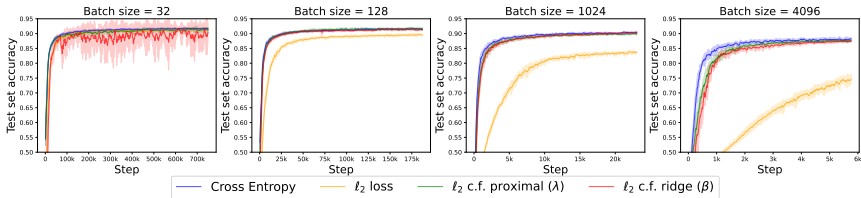

Figure 4: **CIFAR-10 results**. X-axis is the number of iterations, Y-axis is a test set accuracy. Each column corresponds to a different batch size. Different colors indicate different methods.

This performance gap becomes larger as the batch size increases. In CIFAR-10, "$\ell_2$ *c.f. ridge* $(\beta)$" performs similarly to "$\ell_2$ *c.f. proximal* $(\lambda)$", while in CIFAR-100 the method "$\ell_2$ *c.f. ridge* $(\beta)$" fails for small batch sizes. This highlights the impact of the proximal term in our approach which helps avoid overfitting to every batch. For large batch size, both methods perform similarly. Surprisingly, we found that "$\ell_2$ *c.f. proximal* $(\lambda)$" outperformed *Cross Entropy* on CIFAR-100. This finding however does not hold in a larger scale regime on ImageNet as we observe below.

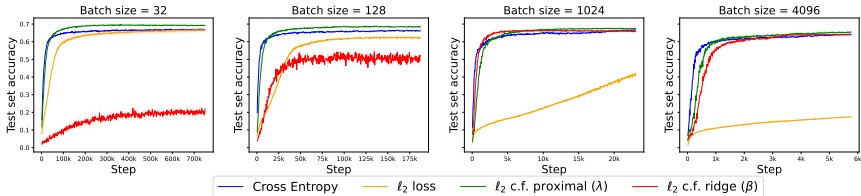

Figure 5: **CIFAR-100 results**. X-axis is the number of iterations, Y-axis is a test set accuracy. Each column corresponds to a different batch size. Different colors indicate different methods.

**Impact of $\lambda$ and $\beta$.** In Figure 6, we report performance at the end of the training as a function $\lambda$ and $\beta$, as well as the best learning rate $\alpha$ for every batch size. For the first two plots we used the learning rates reported in the third plot. The method "$\ell_2$ *c.f. proximal* $(\lambda)$" is overall robust to $\lambda$ provided it is large enough. We only see some sensitivity for smaller batch sizes. The approach "$\ell_2$ *c.f. ridge* $(\beta)$" is more sensitive to the parameter $\beta$ and works better for larger batch sizes. Finally, both of the approaches benefit from large learning rates whenever batch size is increased, while "*Cross Entropy*" and "$\ell_2$ *loss*" require small learning rates.

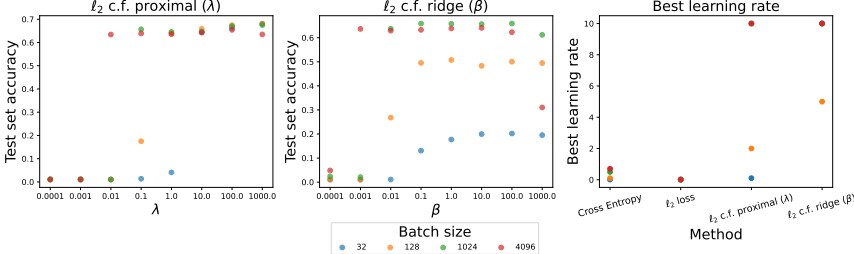

Figure 6: **Dependence on hyperparameters on CIFAR-100**. X-axis is the number of iterations, Y-axis is a test set accuracy. **Left**, ablation over $\lambda$ for '$\ell_2$ *c.f. proximal* $(\lambda)$". **Center**, ablation over $\beta$ for "$\ell_2$ *c.f. ridge* $(\beta)$". **Right**, the best learning rate per method.

**Choice of the algorithm.** We compare the performance of Algorithm 1 and Algorithm 2 in Fig. 7. We see that Algorithm 1 overall leads to better performance than Algorithm 2. Since in Algorithm 2, the backbone is updated using the last layer from the same batch, we hypothesize that this leads to more correlated updates and which may under-perform, while in Algorithm 1, we use the last layer from the previous batch. This motivates the use of this approach.

**Additional ablations.** We ran an ablation on design choices for our method, see Appendix F.1. We only provide a short summary here. We first verified that the inclusion of a bias term in the last

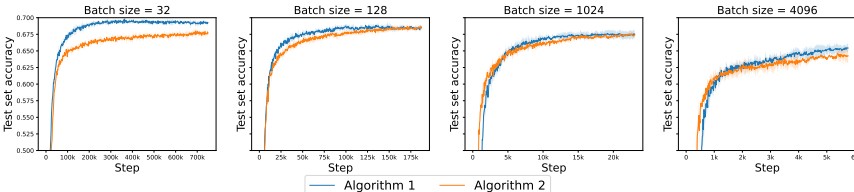

Figure 7: **Comparison of Algorithm 2 and Algorithm 1**. X-axis is the number of iterations, Y-axis is a test set accuracy. A column indicates a batch size while a color represents an algorithm.

layer did not lead to a difference in performance (see Figure F.1). Further, we found that the *zeros* initialization strategy led to the best results (see Figure F.2). Finally, we also saw that using Adam for the backbone performed worse than SGD (see Figure F.3). The Adam update keeps running averages over the gradients and squared gradients which are used to rescale parameter updates. While using momentum over gradients in the backbone works well with our method, the additional step-size rescaling might require us to incorporate an adaptive strategy over $\lambda$ parameter and extend these per last layer dimension, for use with Adam.

**Large scale classification on ImageNet.** We study the performance of our approach on ImageNet. We use NF-Nets-F0 architecture (Brock et al., 2021) with batch size $4096$ and the same training regime as in (Brock et al., 2021). We used $1$ seed for these experiments. See Appendix E for more details. The results are given in Figure 8. We see that our method achieves better performance than "$\ell_2$ *loss*" loss but under-performs "*Cross Entropy*". While this is contrary to our finding on CIFAR-100, ImageNet has ten times the number of classes, so is a significantly different regime. The under-performance of the methods based on the squared loss could be due to advantageous properties of the cross entropy loss in classification, or simply that the training practices with cross entropy have been greatly perfected over the years.

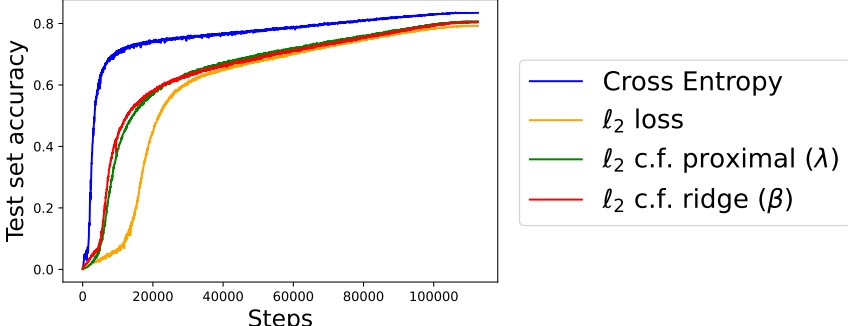

Figure 8: **ImageNet results**. X-axis is the number of iterations, Y-axis is a test set accuracy. Each column corresponds to a different batch size. Different colors indicate different methods.

## 7 CONCLUSION

We have proposed to leverage closed-form optimal solutions for the last layer of neural networks under squared loss throughout optimization. We observe that this accelerates training compared to SGD on squared loss, outperforming SGD on regression tasks and yielding comparable speed to SGD on cross entropy loss on tasks with small-to-moderate number of classes. Regression results are thus particularly promising.

In future work, we will focus on adapting a similar closed-form strategy to the cross entropy loss in the classification setting. We also aim to apply our proximal method to larger scale two-stage settings than DFIV, such as offline reinforcement learning (Chen et al., 2022b) and proxy variables regression (Xu et al., 2021b). Moreover, understanding how to define parameters $\lambda$ per last layer dimension and adapt these over the course of the training is of interest, since it could lead to a better performance with the Adam algorithm.

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

## A  KALMAN FILTER INTERPRETATION OF THE PROXIMAL ALGORITHM

We can interpret the updates on the last layer in Eq. (17) as a Kalman filter under several simplifying assumptions. We take the Bayesian point of view, where we treat the last layer $W_t$ at time $t$ given the feature parameters $\theta_t$ as a random variable. See also (Titsias et al., 2024) for a similar discussion.

First, we assume that the model fits the data perfectly during optimization, so we have the likelihood

$$p(y_i \mid x_i, W_t, \theta_t) = \mathcal{N}(y_i \mid W_t \phi_{\theta_t}(x_i), \sigma_Y^2 I) \tag{22}$$

where $I$ is the identity matrix and $\sigma_Y^2$ is some hyperparameter controlling the variance of the outputs and $(x_i, y_i) \in \mathcal{B}_t$.

Next, we assume $W_t$ evolves like a random walk with Gaussian steps, as the parameters evolve through time $\theta_t$, so

$$p(W_{t+1} \mid W_t) = \mathcal{N}(W_{t+1} \mid W_t, \sigma_W^2 I) \tag{23}$$

where $\sigma_W^2$ is some hyperparameter controlling the variance of the steps.

Equations (22) and (23) provide us a way to update our belief about $W_t$ in closed-form through time. Namely our belief about $W_t$ given our observations $\mathcal{B}_s$ for $s < t$ will be a Gaussian distribution $\mathcal{N}(W_t, \Sigma_t)$, obtained by *Kalman filtering* (Särkkä, 2013). Here, note that $\Sigma_t$ will be a $od \times od$ matrix, i.e. will be squared times the number of parameters in the last layer. For large last layers, this can be intensive to store and manipulate.

Instead, we propose an additional simplifying assumption, where we approximate the dynamics at each step

$$p(W_{t+1} \mid W_t) \approx p(W_{t+1} \mid W_t) \approx \mathcal{N}(W_{t+1} \mid W_t, \sigma_W^2 I). \tag{24}$$

In other words, we ignore the covariance $\Sigma_t$ at each step, and collapse our belief over $W_t$ to the point estimate $W_t$. The resulting update on our point estimate of the last layer is given by maximum-a-posteriori estimation:

$$
\begin{aligned}
W_{t+1} &= \underset{W \in \mathbb{R}^{o \times d}}{\arg\min} - \sum_{(x_i, y_i) \in \mathcal{B}_t} \log p(y_i \mid x_i, W, \theta_t) - \log p(W \mid W_t) \\
&= \underset{W \in \mathbb{R}^{o \times d}}{\arg\min} - \sum_{(x_i, y_i) \in \mathcal{B}_t} \log \mathcal{N}(y_i \mid W_t \phi_{\theta_t}(x_i), \sigma_Y^2 I) - \log \mathcal{N}(W_{t+1} \mid W_t, \sigma_W^2 I) \\
&= \underset{W \in \mathbb{R}^{o \times d}}{\arg\min} \sum_{(x_i, y_i) \in \mathcal{B}_t} \frac{1}{2\sigma_Y^2} \|y_i - W \phi_\theta(x_i)\|_2^2 + \frac{1}{2\sigma_W^2} \|W - W_t\|_F^2 \\
&= W_{\mathcal{B}_t, W_t}^\star(\theta)
\end{aligned}
\tag{25}
$$

with $\lambda = \frac{\sigma_Y^2}{\sigma_W^2}$ in Eq. (13). That is, such approximate Bayesian updates recovers precisely the minimum of the proximal loss Eq. (13), leading to the updates on $W_t$ as in Eqs. (15) and (17).

## B  ALTERNATIVE ALGORITHM

We present an alternative algorithm in Algorithm 2.

---

**Algorithm 2** Proximal closed-form SGD

---

1: **Given:** Batch size $B$, proximal coefficient $\lambda > 0$, neural network $\phi_\theta$ with initial parameters $\theta_0$, learning rate $\alpha > 0$, initial last layer parameters $W_0$.
2: $t \leftarrow 0$
3: **Fit last layer on the first batch**
4: $W_1 \leftarrow W^\star_{\mathcal{B}_1, W_0}(\theta_t)$
5: **while** $\theta_t$ has not converged **do**
6:     $t \leftarrow t + 1$
7:     **Update backbone on the current batch** $\mathcal{B}_t$
8:     $\theta_t \leftarrow \theta_{t-1} - \alpha \nabla_\theta \mathcal{L}_{\mathcal{B}_t}(W_t, \theta_{t-1})$
9:     **Update last layer on the next batch** $\mathcal{B}_{t+1}$
10:     $W_{t+1} \leftarrow W^\star_{\mathcal{B}_{t+1}, W_t}(\theta_t)$
11: **Output:** Optimized $(W^\star, \theta^\star)$

---

## C  PROOFS FOR THE THEORETICAL ANALYSIS OF THE LOSS

This appendix contains the proofs for Section 5.

Spelling out the expression for $\mathcal{L}^\star_{\mathcal{F}}$ in matrix form,

$$
\begin{aligned}
\mathcal{L}^\star_{\mathcal{F}}(\phi) &:= \sum_{i=1}^n \|y_i - W^\star_{\mathcal{F}}(\phi)\phi(x_i)\|_2^2 \\
&= \|Y - W^\star_{\mathcal{F}}(\phi)\phi(X)\|_F^2 \\
&= \mathrm{tr}\left( (Y - W^\star_{\mathcal{F}}(\phi)\phi(X))^\top (Y - W^\star_{\mathcal{F}}(\phi)\phi(X)) \right).
\end{aligned}
\tag{26}
$$

where recall, $\phi(X) \in \mathbb{R}^{d \times n}$, $Y \in \mathbb{R}^{o \times n}$, $W^\star_{\mathcal{F}}(\phi) := Y\phi(X)^\top \left(\phi(X)\phi(X)^\top + \beta I\right)^{-1} \in \mathbb{R}^{o \times d}$, and

The derivative of $\mathcal{L}^\star_{\mathcal{F}}$ at $\phi$ is a linear map $\mathrm{D}\mathcal{L}^\star_{\mathcal{F}}(\phi)\colon \mathcal{F} \to \mathbb{R}$. Just as in Theorem 1, to calculate $\mathrm{D}\mathcal{L}^\star_{\mathcal{F}}(\phi)$ we do not need to differentiate $W^\star_{\mathcal{F}}(\phi)$ with respect to $\phi$, and may treat it as constant instead. So for $\psi \in \mathcal{F}$,

$$
\mathrm{D}\mathcal{L}^\star_{\mathcal{F}}(\phi)[\psi] = \mathrm{tr}\Big( \underbrace{-2(Y - W^\star_{\mathcal{F}}(\phi)\phi(X))^\top W^\star_{\mathcal{F}}(\phi)}_{=: \nabla \mathcal{L}^\star_{\mathcal{F}}(\phi)^\top} \psi(X) \Big).
\tag{27}
$$

The definition of the gradient $\nabla \mathcal{L}^\star_{\mathcal{F}}(\phi) \in \mathbb{R}^{o \times n}$ is just a standalone definition used for convenience as – without an inner product on $\mathcal{F}$ – we do not have a well defined notion of gradients for the functional $\mathcal{L}^\star_{\mathcal{F}}$. Importantly, note that $\phi^\star$ is a critical point of $\mathcal{L}^\star_{\mathcal{F}}$ if and only if $\mathrm{D}\mathcal{L}^\star_{\mathcal{F}}(\phi^\star) = 0$, i.e. if and only if $\mathrm{D}\mathcal{L}^\star_{\mathcal{F}}(\phi^\star)[\psi] = 0$ for all $\psi \in \mathcal{F}$, i.e. if and only if $\nabla \mathcal{L}^\star_{\mathcal{F}}(\phi) = 0$.

Plugging in the expression for $W^\star_{\mathcal{F}}(\phi)$ in the definition of $\nabla \mathcal{L}^\star_{\mathcal{F}}(\phi)$ and writing $\Phi := \phi(X) \in \mathbb{R}^{d \times n}$, we get

$$
\begin{aligned}
\nabla \mathcal{L}^\star_{\mathcal{F}}(\phi) &= 2 W^\star_{\mathcal{F}}(\phi)^\top (W^\star_{\mathcal{F}}(\phi)\phi(X) - Y) \\
&= 2(\Phi\Phi^\top + \beta I)^{-1}\Phi Y^\top Y(\Phi^\top(\Phi\Phi^\top + \beta I)^{-1}\Phi - I).
\end{aligned}
\tag{28}
$$

Define

$$
Y_\star = Y\Phi^\top(\Phi\Phi^\top + \beta I)^{-1}\Phi, \quad Y_\perp = Y(I - \Phi^\top(\Phi\Phi^\top + \beta I)^{-1}\Phi).
\tag{29}
$$

In particular

$$
Y = Y_\star + Y_\perp.
\tag{30}
$$

$Y_\star$ should be thought of the part of the outputs "attainable" by the features $\Phi$, and $Y_\perp$ the part of the outputs "unattainable". To see this, take a singular value decomposition of $\Phi$ of the form

$\Phi = U\Sigma V^\top$ where $U \in \mathbb{R}^{d \times d}$ has orthonormal columns, $\Sigma \in \mathbb{R}^{d \times d}$ is diagonal with the first $r := \operatorname{rank} \Phi$ diagonal entries being non-zero and $V \in \mathbb{R}^{n \times d}$ has orthonormal columns. Moreover write $\mathbb{V}_\star \leq \mathbb{R}^n$ for the subspace spanned by the rows of $\Phi$, or equivalently the first $r$ columns of $V$, the space of "attainable" outputs. Further let $\mathbb{V}_\perp \leq \mathbb{R}^n$ its orthogonal complement, the space of "unattainable" outputs. Observe that

$$\Phi^\top (\Phi\Phi^\top + \beta I)^{-1}\Phi = V\Sigma(\Sigma^2 + \beta I)^{-1}\Sigma V^\top. \tag{31}$$

When acting on the right, its image is $\mathbb{V}_\star$. Therefore the rows of $Y_\star$ are in $\mathbb{V}_\star$, and those of $Y_\perp$ are in $\mathbb{V}_\perp$. In particular, note that when $\beta = 0$ and assuming $\Phi$ has full row rank, $\Phi^\top(\Phi\Phi^\top)^{-1}\Phi$ is the orthogonal projection onto $\mathbb{V}_\star$ when acting on the right.

Now note that

$$(\Phi\Phi^\top + \beta I)^{-1}\Phi Y^\top = U(\Sigma^2 + \beta I)^{-1}\Sigma V^\top Y^\top = U(\Sigma^2 + \beta I)^{-1}\Sigma V^\top Y_\star^\top = (\Phi\Phi^\top + \beta I)^{-1}\Phi Y_\star^\top. \tag{32}$$

So from Eq. (28),

$$\nabla \mathcal{L}_{\mathcal{F}}^\star(\phi) = -2(\Phi\Phi^\top + \beta I)^{-1}\Phi Y_\star^\top Y_\perp. \tag{33}$$

And applying $\Phi^\top$ on the left we see that $\nabla \mathcal{L}_{\mathcal{F}}^\star(\phi) = 0$ if and only if $\Phi^\top \nabla \mathcal{L}_{\mathcal{F}}^\star(\phi) = 0$, i.e. $Y_\star^\top Y_\perp = 0$.

We see for example that $\phi = 0$ gives $Y_\star = 0$ and $Y_\perp = Y$, so is a critical point of $\mathcal{L}_{\mathcal{F}}^\star$, even though it is not a global minimizer, assuming $Y \neq 0$. When $\min(d, n') \geq \operatorname{rank} Y$, where $n'$ is the number of distinct $x_i$, being a global minimizer is equivalent to $Y_\perp = 0$, because then the features can "attain" $Y$.

When $Y \neq 0$, $\mathcal{L}_{\mathcal{F}}^\star$ admits critical points which are not global minima thus it is not convex, so this concludes the proof of Theorem 3.

Remark that a result such as Theorem 1 cannot be extended to second derivatives, otherwise we could differentiate $\mathcal{L}_{\mathcal{F}}^\star$ twice by keeping $W_{\mathcal{F}}^\star(\phi)$ constant, and would obtain that the Hessian of $\mathcal{L}_{\mathcal{F}}^\star$ is positive semi-definite since so is the one of the squared loss. But this is impossible since we showed that $\mathcal{L}_{\mathcal{F}}^\star$ is not convex.

### C.1 NEURAL TANGENT KERNEL INFINITE WIDTH LIMIT

Before proving Theorem 4, we provide a self-contained overview of the neural tangent kernel (NTK) limit, based on Jacot et al. (2018).

$\phi_\theta$ is assumed to be a neural network, $\theta$ are its parameters consisting of weights $W^{(\ell)}$ and biases $b^{(\ell)}$, such that $\phi_\theta(x) = \alpha_\theta^{(L)}(x)$, the pre-activations $\tilde{\alpha}_\theta^{(\ell)}(x) \colon \mathbb{R}^{d_0} \to \mathbb{R}^{d_\ell}$ the activations $\alpha_\theta^{(\ell)}(x) \colon \mathbb{R}^{d_0} \to \mathbb{R}^{d_\ell}$, $d_L = d$ and

$$\alpha_\theta^{(0)}(x) = x$$
$$\tilde{\alpha}_\theta^{(\ell+1)}(x) = \frac{1}{\sqrt{d_\ell}} W^{(\ell)} \alpha_\theta^{(\ell)}(x) + b^{(\ell)} \tag{34}$$
$$\alpha_\theta^{(\ell)}(x) = \sigma\left(\tilde{\alpha}_\theta^{(\ell)}(x)\right),$$

where $\sigma \colon \mathbb{R} \to \mathbb{R}$ is a twice differentiable non-linearity function with bounded second derivative, applied element-wise. The parameters $\theta$ are initialized with $W_{ij}^{(\ell)} \sim \mathcal{N}(0, 1)$, $b_j^{(\ell)} \sim \mathcal{N}(0, 1)$ which, combined with the pre-multiplicative factors $1/\sqrt{d_\ell}$ in Eq. (34), corresponds to the LeCun initialization (see Section 4.1).

We then consider gradient flow on the loss $\mathcal{L}^\star$:

$$\frac{\mathrm{d}\theta}{\mathrm{d}t} = -\nabla_\theta \mathcal{L}^\star(\theta). \tag{35}$$

We further consider the infinite width limit $n_1, \ldots, n_L \to \infty$ sequentially, that is we first take $n_1 \to \infty$, then $n_2 \to \infty$, etc. In this limit, the dynamics of the function $\phi = \phi_\theta$ under Eq. (35) are given by kernel gradient descent: for $x \in \mathbb{R}^{d_0}$,

$$\frac{\mathrm{d}\phi(x)}{\mathrm{d}t} = -K(x, X)\nabla \mathcal{L}_{\mathcal{F}}^\star(\phi)$$
$$= 2K(x, X)(\Phi\Phi^\top + \beta I)^{-1}\Phi Y_\star^\top Y_\perp \tag{36}$$

where we used Eq. (33), and $K$ is a positive semi-definite kernel $K \colon \mathbb{R}^{d_0} \times \mathbb{R}^{d_0} \to \mathbb{R}^{d \times d}$, the *neural tangent kernel* (Jacot et al., 2018, Theorem 2). Whenever this kernel is positive definite (see for example Jacot et al. (2018, Proposition 2)) we know that, as $t \to \infty$, $\phi$ will converge pointwise to a critical point $\phi^\star$ of $\mathcal{L}_{\mathcal{F}}^\star$. The goal of Theorem 4 is to show that, almost surely in the initialization, $\phi^\star$ will be a global minimum of $\mathcal{L}_{\mathcal{F}}^\star$. In other words, $Y_\star \to Y$ and $Y_\perp \to 0$ as $t \to \infty$.

From Eq. (36), we see that

$$\frac{\mathrm{d}\Phi}{\mathrm{d}t} = 2\Xi(\Phi\Phi^\top + \beta I)^{-1}\Phi Y_\star^\top Y_\perp \tag{37}$$

where $\Xi := K(X, X) \in \mathbb{R}^{d \times d}$. So

$$
\begin{aligned}
\frac{\mathrm{d}Y_\star}{\mathrm{d}t} &= \frac{\mathrm{d}}{\mathrm{d}t}\left(Y\Phi^\top(\Phi\Phi^\top + \beta I)^{-1}\Phi\right) \\
&= Y\frac{\mathrm{d}\Phi^\top}{\mathrm{d}t}(\Phi\Phi^\top + \beta I)^{-1}\Phi - Y\Phi^\top(\Phi\Phi^\top + \beta I)^{-1}\frac{\mathrm{d}\Phi}{\mathrm{d}t}\Phi^\top(\Phi\Phi^\top + \beta I)^{-1}\Phi \\
&\quad - Y\Phi^\top(\Phi\Phi^\top + \beta I)^{-1}\Phi\frac{\mathrm{d}\Phi^\top}{\mathrm{d}t}(\Phi\Phi^\top + \beta I)^{-1}\Phi + Y\Phi^\top(\Phi\Phi^\top + \beta I)^{-1}\frac{\mathrm{d}\Phi}{\mathrm{d}t} \\
&= 2YY_\perp^\top Y_\star\Phi^\top(\Phi\Phi^\top + \beta I)^{-1}\Xi(\Phi\Phi^\top + \beta I)^{-1}\Phi \\
&\quad - 2Y\Phi^\top(\Phi\Phi^\top + \beta I)^{-1}\Xi(\Phi\Phi^\top + \beta I)^{-1}\Phi Y_\star^\top \underbrace{Y_\perp\Phi^\top(\Phi\Phi^\top + \beta I)^{-1}\Phi}_{=0} \\
&\quad - 2Y\underbrace{\Phi^\top(\Phi\Phi^\top + \beta I)^{-1}\Phi Y_\perp^\top}_{=0}Y_\star\Phi^\top(\Phi\Phi^\top + \beta I)^{-1}\Xi(\Phi\Phi^\top + \beta I)^{-1}\Phi \\
&\quad + 2Y\Phi^\top(\Phi\Phi^\top + \beta I)^{-1}\Xi(\Phi\Phi^\top + \beta I)^{-1}\Phi Y_\star^\top Y_\perp \\
&= 2Y_\perp Y_\perp^\top Y_\star\Phi^\top(\Phi\Phi^\top + \beta I)^{-1}\Xi(\Phi\Phi^\top + \beta I)^{-1}\Phi \\
&\quad + 2Y_\star\Phi^\top(\Phi\Phi^\top + \beta I)^{-1}\Xi(\Phi\Phi^\top + \beta I)^{-1}\Phi Y_\star^\top Y_\perp.
\end{aligned}
\tag{38}
$$

Hence

$$
\begin{aligned}
\frac{\mathrm{d}Y_\star}{\mathrm{d}t}Y_\star^\top &= \frac{\mathrm{d}Y_\star}{\mathrm{d}t}Y_\star^\top + Y_\star\frac{\mathrm{d}Y_\star^\top}{\mathrm{d}t} \\
&= 2Y_\perp Y_\perp^\top Y_\star\Phi^\top(\Phi\Phi^\top + \beta I)^{-1}\Xi(\Phi\Phi^\top + \beta I)^{-1}\Phi Y_\star^\top \\
&\quad + 2Y_\star\Phi^\top(\Phi\Phi^\top + \beta I)^{-1}\Xi(\Phi\Phi^\top + \beta I)^{-1}\Phi Y_\star^\top \underbrace{Y_\perp Y_\star^\top}_{=0} \\
&= 2\underbrace{\left(Y_\perp Y_\perp^\top\right)}_{=:A}\underbrace{\left(Y_\star\Phi^\top(\Phi\Phi^\top + \beta I)^{-1}\Xi(\Phi\Phi^\top + \beta I)^{-1}\Phi Y_\star^\top\right)}_{=:B}.
\end{aligned}
\tag{39}
$$

So

$$\frac{\mathrm{d}}{\mathrm{d}t}\left(Y_\star Y_\star^\top\right) = \frac{\mathrm{d}Y_\star}{\mathrm{d}t}Y_\star^\top + Y_\star\frac{\mathrm{d}Y_\star^\top}{\mathrm{d}t} = 2(AB + BA). \tag{40}$$

We can assume without loss of generality that $\operatorname{rank} Y = o$; linearly related rows of $Y$ will induce linearly related rows of $Y_\star$ and $Y_\perp$ at all time, so for the sake of analysis we may ignore linearly dependent rows. We also assume without loss of generality that the $x_i$ are distinct, otherwise we may ignore the corresponding duplicate columns of $\Phi$.

At initialization, the infinite width neural network is a Gaussian process (Jacot et al., 2018, Proposition 1). If the corresponding *neural Gaussian process kernel* (NGPK) is positive definite (see for example Gao et al. (2023, Theorem 4.5)), and if all $x_i$ are distinct, then the distribution of the columns of $\Phi$ follow a non-degenerate Gaussian at initialization ($t = 0$). So, at $t = 0$, $\dim \mathbb{V}_\star = \operatorname{rank} \Phi = \min(d, n)$ almost surely. Since $\min(d, n) \geq \operatorname{rank} Y = o$, projecting the rows of $Y$ onto $\mathbb{V}_\star$ we get that $\operatorname{rank} Y_\star = o$ almost surely at $t = 0$, so B is positive definite almost surely.

$A$ is positive semi-definite so, by Eq. (40), $\frac{\mathrm{d}}{\mathrm{d}t}\left(Y_\star Y_\star^\top\right)$ is positive semi-definite. So the (almost surely) positive eigenvalues of $Y_\star Y_\star^\top$ are non-decreasing through time. They converge when $A = 0$,

i.e. $Y_\perp = 0$, which corresponds to a global minimum of $\mathcal{L}_\mathcal{F}^\star$. They are guaranteed to converge when the NTK is positive definite.

We see in addition that, since the NTK is the sum of the NGPK with some other positive semi-definite kernel (Jacot et al., 2018, Theorem 1), the NGPK being positive definite implies that the NTK is too. So this concludes the proof of Theorem 4.

## D  DEEP FEATURE INSTRUMENTAL VARIABLE REGRESSION

In Instrumental Variable Regression, we observe a treatment $X$ and an outcome $Y$. But we have an unobserved confounder that affects both $X$ and $Y$, specifically we have the relation

$$Y = f_{\text{struct}}(X) + \epsilon, \quad \mathbb{E}[\epsilon] = 0, \quad \mathbb{E}[\epsilon \mid X] \neq 0 \tag{41}$$

where $f_{\text{struct}}$ is called the structural function which we aim to infer, and $\epsilon$ is an additive noise term. Because $\mathbb{E}[\epsilon \mid X] \neq 0$, we cannot use ordinary supervised learning techniques. Instead we assume we have access to an instrumental variable $Z$ which satisfies $\mathbb{E}[\epsilon \mid Z] = 0$. Then we have that $\mathbb{E}[Y \mid Z] = \mathbb{E}[f_{\text{struct}}(X) \mid Z]$, so we solve this equation for $f_{\text{struct}}$.

Deep Feature Instrumental Variable Regression (DFIV) (Xu et al., 2020) solves this by using two neural networks. The first neural network models $w^\top \psi_{\theta_X}(x) = f_{\text{struct}}(x)$, and the second neural network models $W\phi_{\theta_Z}(z) = \mathbb{E}[\psi_{\theta_X}(X) \mid Z = z]$. It alternates between two stages. In the first stage $W$ and $\theta_Z$ are regressed to fit

$$W\phi_{\theta_Z}(z) = \mathbb{E}[\psi_{\theta_X}(X) \mid Z = z]. \tag{42}$$

using a squared loss on some data $\{(x_i^{(1)}, z_i^{(1)})\}$:

$$\mathcal{L}^{(1)}(W, \theta_Z) := \sum_i \|W\phi_{\theta_Z}(z_i^{(1)}) - \psi_{\theta_X}(x_i^{(1)})\|_2^2 + \text{regularizer}(W) \tag{43}$$

Solving $W$ in closed-form with a ridge or proximal regularizer makes it implicitly depend on $\theta_X$, which we write $W^\star(\theta_X)$. Leveraging this dependence, in the second stage $w$ and $\theta_X$ are regressed to fit

$$w^\top W^\star(\theta_X)\phi_{\theta_Z}(z) = \mathbb{E}[Y \mid Z = z] \tag{44}$$

using a squared loss on some data $\{(y_i^{(2)}, z_i^{(2)})\}$:

$$\mathcal{L}^{(2)}(w, \theta_X) := \sum_i \|w^\top W^\star(\theta_X)\phi_{\theta_Z}(z_i^{(2)}) - y_i^{(2)}\|_2^2 + \text{regularizer}(W). \tag{45}$$

When both Eqs. (42) and (44) are simultaneously satisfied we see that $\mathbb{E}[w^\top \psi_{\theta_X}(X) \mid Z] = \mathbb{E}[Y \mid Z] = \mathbb{E}[f_{\text{struct}}(X) \mid Z]$, as required. Since this is a bilevel optimization problem, we alternate between the two stages. In both stages we use either "$\ell_2$ *c.f. proximal* $(\lambda)$" or the original method, which relies on "$\ell_2$ *c.f. ridge* $(\beta)$" together with backpropagation through the closed-form solution. Our "$\ell_2$ *c.f. proximal* $(\lambda)$" is thus much cheaper. With "$\ell_2$ *c.f. proximal* $(\lambda)$", we use three distinct proximal hyperparameters $\lambda$: one hyperparameter $\lambda_1$ for the closed-form solution of $W$ in stage 1, one hyperparameter $\lambda_2$ for the closed-form solution of $w$ in stage 2, and one hyperparameter $\lambda_{1,2}$ for the closed-form solution of $W$ in stage 2. This last step is performed before updating $\theta_Z$ and $w$ to obtain the closed-form solution $W^\star(\theta_X)$ as a function of $\theta_X$. See Algorithm 3 for details.

## E  EXPERIMENTAL DETAILS

**DFIV regression.**  For the experiments, we follow closely (Xu et al., 2020) and we consider a slightly modified version of `d-spirtes` task (Matthey et al., 2017). This is an image dataset described by five latent parameters (`shape, scale, rotation, posX, posY`). The images are $64 \times 64 = 4096$ dimensional. In this experiment, the authors fix the `shape` parameter to `heart`, i.e., they only used heart-shaped images. The authors generated data for IV regression in which they use each figure as a treatment variable $X$. Hence, the treatment variable is 4096-dimensional in this experiment. To make the task more challenging, they used `posY` as the hidden confounder, which

---

**Algorithm 3** DFIV proximal

---

1: **Given:** Stage 1 data $\{(x_i^{(1)}, z_i^{(1)})\}$, stage 2 data $\{(y_i^{(2)}, z_i^{(2)})\}$, batch sizes $B_1, B_2$, proximal coefficients $\lambda_1, \lambda_2, \lambda_{1,2} > 0$, neural networks $\psi_{\theta_X}, \phi_{\theta_Z}$ with initial parameters $\theta_{X0}, \theta_{Z0}$ respectively, learning rates $\alpha_1, \alpha_2 > 0$, initial last layer parameters $w_0, W_0$, number of updates in each stage $T_1, T_2$.
2: $t_1 \leftarrow 0$
3: $t_2 \leftarrow 0$
4: **while** $\theta_{Zt_1}$ and $\theta_{Xt_2}$ have not converged **do**
5:     Sample $B_1$ stage 1 data $\mathcal{B}^{(1)} \subset \{(x_i^{(1)}, z_i^{(1)})\}$, and $B_2$ stage 2 data $\mathcal{B}^{(2)} \subset \{(y_i^{(2)}, z_i^{(2)})\}$
6:     **for** $t = 1$ to $T_1$ **do**
7:         $t_1 \leftarrow t_1 + 1$
8:         $\theta_{Zt_1} \leftarrow \theta_{Z(t_1-1)} - \alpha_1 \nabla_{\theta_Z} \mathcal{L}_{\mathcal{B}^{(1)}}^{(1)}(W_{t_1-1}, \theta_{Z(t_1-1)})$
9:         $W_{t_1} \leftarrow W_{\mathcal{B}^{(1)}, W_{t_1-1}}^{\star}(\theta_{Zt_1})$ on loss $\mathcal{L}_{\mathcal{B}^{(1)}}^{(1)}$ with proximal coefficient $\lambda_1$
10:     **for** $t = 1$ to $T_2$ **do**
11:         $t_2 \leftarrow t_2 + 1$
12:         $W^{\star}(\theta_X) \leftarrow W_{\mathcal{B}^{(1)}, W_{t_1}}^{\star}(\theta_{Z0})$ on loss $\mathcal{L}_{\mathcal{B}^{(1)}}^{(1)}$ with proximal coefficient $\lambda_{1,2}$
13:         $\theta_{Xt_2} \leftarrow \theta_{X(t_2-1)} - \alpha_2 \nabla_{\theta_X} \mathcal{L}_{\mathcal{B}^{(2)}}^{(2)}(w_{t_2-1}, \theta_{X(t_2-1)})$
14:         $w_{t_2} \leftarrow w_{\mathcal{B}^{(2)}, w_{t_2-1}}^{\star}(\theta_{Xt_2})$ on loss $\mathcal{L}_{\mathcal{B}^{(2)}}^{(2)}$ with proximal coefficient $\lambda_2$
15: **Output:** Optimized $(w^{\star}, W^{\star}, \theta_X^{\star}, \theta_Z^{\star})$

---

is not revealed to the model. They used three latent varaibles as the instrument variables $Z$. The outcome $Y$ is defined as

$$Y = f_{\text{struct}}(X) + 32(\mathtt{pos}\mathtt{Y} - 0.5) + \epsilon, \tag{46}$$

where $\epsilon \sim \mathcal{N}(0, 0.5)$. Here, we used $f_{\text{struct}}(X)$ from a different paper (Xu et al., 2021a), which was defined as

$$f_{\text{struct}}(X) = \frac{(\mathtt{vec}(B)^{\top}X)^2 - 3000}{500}, \tag{47}$$

where $B \in \mathbb{R}^{64 \times 64}$, $B_{ij} = \frac{|32-j|}{32}$ and $\mathtt{vec}(B)$ collapses the matrix $B$ to a vector of dimensionality 4096. The choice of this structural function was motivated by (Xu et al., 2021a), because the original choice described in Xu et al. (2020) led to essentially a constant function (in expectation).

For our experiments, we use different batch sizes. The DFIV method (Xu et al., 2020) essentially corresponds to two-stage "$\ell_2$ *c.f. ridge* ($\beta$)" where we have $\beta_1$ and $\beta_2$ parameters for the first and second stage correspondingly. In our proximal method, DFIV proximal, as described in Appendix D, we have three parameters $\lambda_1$, $\lambda_2$ for the first and second stage proximal updates and $\lambda_{1,2}$ for first-stage update inside the second stage. In practice, we sweep over $\lambda_1$ and $\lambda_2$ and we use $\lambda_{1,2}$ to be very small, i.e. $\lambda_{1,2} = 0.0001$ as we found that using large $\lambda_{1,2}$ did not work well. We choose $T_1 = 20$ and $T_2 = 1$ as in Xu et al. (2020).

The datasets are split into the training set with 10000 points, validation set with 100 points and holdout test set with 488 points.

When we evaluate performance, we use two strategies. One is following (Xu et al., 2020) and whenever performance is reported, takes the first stage and second stage backbone parameters, and re-estimates the corresponding last layers on the whole 10000 training set. In Figure 3 it is represented by the solid line. The second strategy just takes the current estimates of the last layers. In Figure 3 it is represented by the dashed line.

The sweep range for $\beta_1, \beta_2, \lambda_1, \lambda_2$ is $\{0.00001, 0.0001, 0.001, 0.01, 0.1, 1.0, 10.0, 100.0, 1000.0, 10000.0\}$. On top of that, we also sweep over the learning rate (we use the same learning rate for both stages) in the range $\{0.001, 0.005, 0.01, 0.05, 0.1\}$. Each experiment is run with 3 seeds. The best hyperparameters are selected by minimizing the mean squred error (MSE) on the validation set at the end of the training, using the first evaluation strategy (re-estimating the last layers on the whole dataset). The performance is reported on the holdout test set.

**CIFAR-10, CIFAR-100.** We always use SGD optimizer with Nesterov momentum $\gamma = 0.9$. We train on the $80\%$ of the training set and we use the remaining $20\%$ for validation. For reporting performance, we use the corresponding test set. The sweep ranges are $\alpha \in \{10.0, 5.0, 2.0, 1.0, 0.5, 0.3, 0.2, 0.1, 0.05, 0.01, 0.005, 0.001, 0.0005\}$, learning rate and $\lambda \in \{0.0001, 0.001, 0.01, 0.1, 1.0, 10.0, 100.0, 1000.0]$, $\beta \in \{0.0001, 0.001, 0.01, 0.1, 1.0, 10.0, 100.0, 1000.0\}$.

**ImageNet.** We follow closely the experimental setup described in (Brock et al., 2021), including learning rate schedule, label smoothing, data augmentations and Nesterov momentum in the SGD. The learning schedule is a warmup cosine decay with the peak learning rate $\alpha = 1.6$. We also swept over $\alpha \in [0.1, 1., 1.6, 2., 5.0]$ range. Overall, all the methods performed the best with $\alpha = 1.6$ except for "$\ell_2$ *loss*" which performed the best with $\alpha = 1$. For "$\ell_2$ *c.f. proximal ($\lambda$)*", we used $\lambda = 10000$ and for "$\ell_2$ *c.f. ridge ($\beta$)*", we used $\beta = 0.01$. To select these parameters, we ran a sweep over $\beta \in [0.0001, 0.001, 0.01, 0.1, 1.0, 10.0, 100.0, 1000.0, 10000.0]$ and over $\lambda \in [0.0001, 0.001, 0.01, 0.1, 1.0, 10.0, 100.0, 1000.0, 10000.0]$. We used validation set of ImageNet for the hyperparmaeters selection.

# F    ADDITIONAL RESULTS

## F.1    HYPERPARAMETER ABLATIONS ON CIFAR-100.

In this section, we provide results for ablating hyperparameters and design choices for our method.

**Use of a bias.** We study the impact of using bias on the performance of "$\ell_2$ *c.f. proximal ($\lambda$)*" on CIFAR-100. The results are given in Figure F.1. We observe similar performance for both strategies.

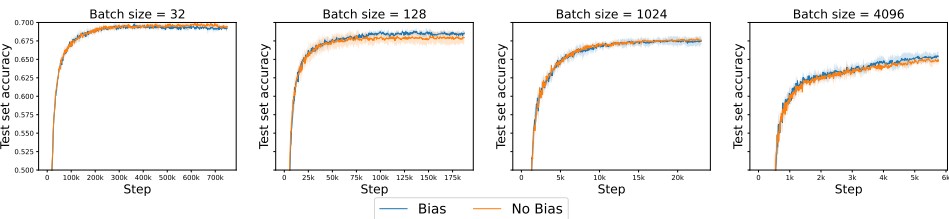

Figure F.1: **Whether to use a bias**. X-axis is the number of iterations, Y-axis is a test set accuracy. Each column corresponds to a different batch size. Different colors indicate different methods.

**Impact of initialization.** We study impact of different initialization strategies (see Section 4.1 for more details). The results are given in Figure F.2. We see that using *zero* initialization leads to overall better performance across batch sizes.

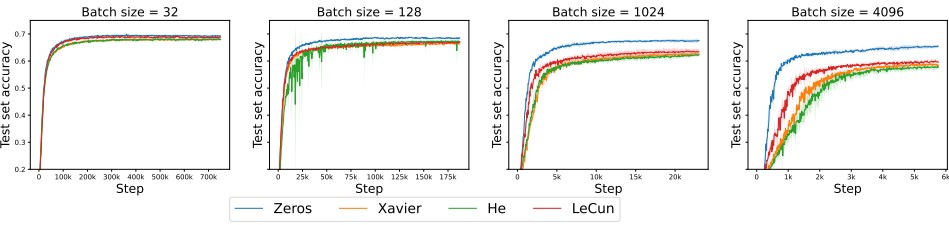

Figure F.2: **Initialization strategy**. X-axis is the number of iterations, Y-axis is a test set accuracy. Each column corresponds to a different batch size. Different colors indicate different methods.

**Adam optimizer.** We train our method "$\ell_2$ *c.f. proximal ($\lambda$)*" but replacing the SGD update on $\theta$ in Eq. (17) by an Adam optimizer update. The results are reported in Fig. F.3. We observe that

using Adam together with our method leads to worse performance. The Adam update keeps running averages over the gradients and squared gradients which are used to rescale parameter updates. While using momentum over gradients in the backbone works well with our method, the additional step-size rescaling might require us to incorporate an adaptive strategy over $\lambda$ parameter and extend these per last layer dimension.

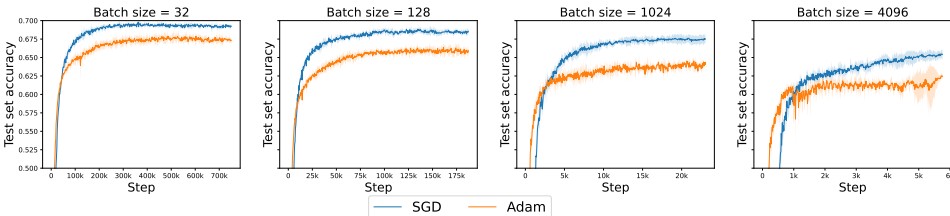

Figure F.3: **Adam vs SGD**. X-axis is the number of iterations, Y-axis is a test set accuracy. Each column corresponds to a different batch size. Different colors indicate different methods.

### F.2 CIFAR-100

We present here the results for Algorithm 2. The summary results on CIFAR-100 are given in Fig. F.4. The bias or no bias ablation is given in Fig. F.5. The SGD vs Adam ablation is given Fig. F.6. The ablation on the initialization strategy is given in Fig. F.7.

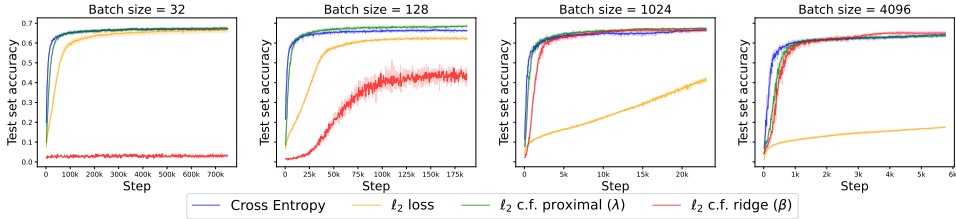

Figure F.4: **CIFAR-100 results, Algorithm 2.**. X-axis is the number of iterations, Y-axis is a test set accuracy. Each column corresponds to a different batch size. Different colors indicate different methods.

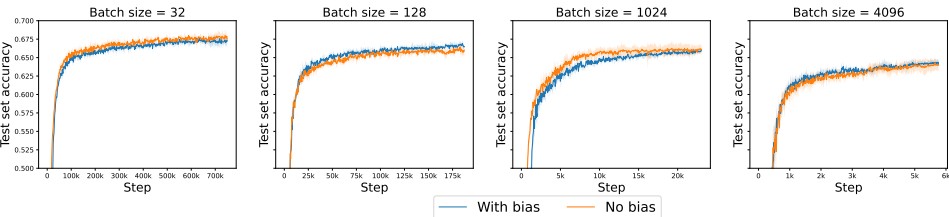

Figure F.5: **Whether to use a bias, Algorithm 2.**. X-axis is the number of iterations, Y-axis is a test set accuracy. Each column corresponds to a different batch size. Different colors indicate different methods.

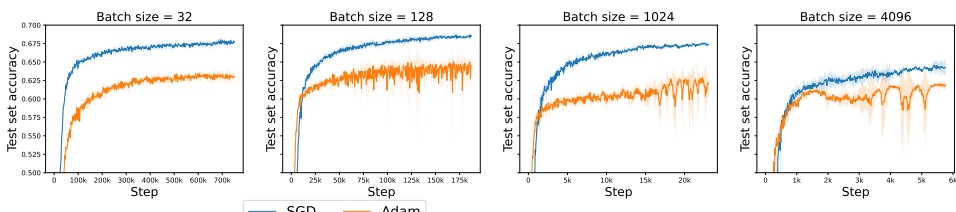

Figure F.6: **Adam vs SGD, Algorithm 2**. X-axis is the number of iterations, Y-axis is a test set accuracy. Each column corresponds to a different batch size. Different colors indicate different methods.

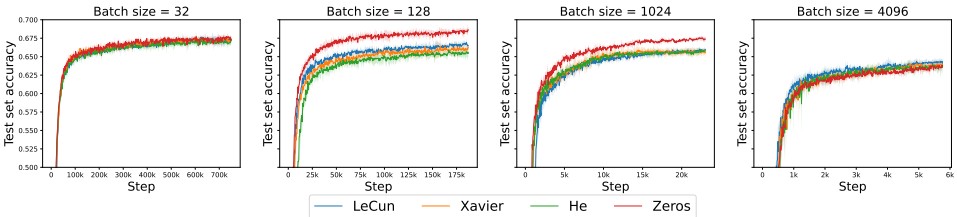

Figure F.7: **Initialization strategy, Algorithm 2**. X-axis is the number of iterations, Y-axis is a test set accuracy. Each column corresponds to a different batch size. Different colors indicate different methods.