# OpenReview forum: "Closed-form Last Layer Optimization"
_ICLR.cc/2026/Conference — Submitted to ICLR 2026_

### Official Review · Reviewer_JUgn · 2025-10-27

**Soundness:** 3
**Presentation:** 3
**Contribution:** 2
**Rating:** 4
**Confidence:** 3

**Summary:**

This paper proposes training with the closed-form last-layer solution (under squared loss): at each step, recompute the final linear weights exactly from current features and update only the backbone by gradient descent. By the envelope theorem, this avoids back-propagating through the matrix inverse and is equivalent to alternating a backbone GD step with an exact least-squares solve for the last layer. For minibatch SGD, the authors add a proximal regularizer that pulls the new last-layer solution toward the previous one, preventing per-batch overfitting; the resulting loop alternates (i) a backbone GD step and (ii) a closed-form ridge solve with proximal coupling, admitting an approximate Kalman filter / RLS interpretation. A backbone-first update order works best empirically.

Theoretically, optimizing the induced implicit loss $L^{*}(\theta)$ is non-convex with possible spurious critical points, but in the infinite-width NTK regime (with a positive-definite kernel and sufficient feature rank) gradient descent converges to a global optimum. Empirically, on regression (FNO/PDEs and DFIV) the method converges faster and attains lower MSE than $\ell\_{2}$-SGD, and in DFIV removes the need for a costly final refit. On CIFAR-10/100, closed-form $\ell\_2$ training consistently beats $\ell\_2$-SGD and can match or exceed cross-entropy on CIFAR-100, though on ImageNet cross-entropy remains stronger. Overall: a simple, practical algorithm with supporting theory and broad empirical gains, especially for squared-loss training.

**Strengths:**

- Novel Optimization Approach: The idea of enforcing the last layer to be optimal at each step is a fresh departure from standard end-to-end SGD. It exploits a known closed-form solution in an innovative way to simplify the optimization problem, essentially performing exact minimization over last-layer weights in each iteration. This two-timescale strategy has appeared in theory, but this work is the first to turn it into a practical training method integrated with SGD.

 - Theoretical Insight: The paper offers non-trivial theoretical contributions. It analyzes the implicit loss landscape when the last layer is always optimal, proving that while this landscape is generally non-convex with potentially bad critical points, gradient descent in the infinite-width NTK limit will avoid those and find a global minimizer. This convergence theorem (Theorem 4) under positive-definite kernel assumptions is a reassuring theoretical justification for the method’s efficacy. Moreover, the use of the envelope theorem to show that backpropagation through the closed-form solution is unnecessary (Theorem 1 and 2) is a nice theoretical simplification that saves computation.

 - Practical Stochastic Algorithm: The paper identifies and tackles the key challenge of mini-batch training (last-layer overfitting to each batch) by introducing a proximal regularization to the closed-form update. This is a simple and effective fix that keeps the last layer update stable and coupled to the backbone’s progress, unlike a naive moving average approach which could decouple and diverge. The resulting algorithm (Algorithm 1) is easy to implement and can plug into existing training pipelines, as it essentially alternates a usual backbone SGD step with solving a small linear system for the last layer.

 - Empirical Performance: Across regression tasks, closed-form last-layer training yields lower MSE and faster convergence than vanilla SGD, with especially strong gains in small-batch settings (the proximal update stabilizes stochasticity). On DFIV, it outperforms small-batch baselines and nearly matches costly two-stage refitting—eliminating that step. In classification, squared-loss with closed-form updates consistently beats $\ell_2$-SGD on CIFAR-10/100, and on CIFAR-100 it even modestly outperforms cross-entropy. Overall, results show both optimization speedups and better generalization in many cases.

**Weaknesses:**

- Restriction to Squared Loss: A notable limitation is that the method inherently relies on the squared loss to obtain a closed-form solution for the last layer. This means it cannot be directly applied to tasks where cross-entropy or other non-quadratic loss functions are standard. In classification, using a squared error surrogate is somewhat non-standard and requires a heuristic at prediction time (taking an argmax of outputs since they don’t form a probability distribution). While the authors show this can work reasonably, it forgoes the probabilistic interpretation and other benefits of the softmax cross-entropy framework. The approach’s success on CIFAR-100 vs. cross-entropy is intriguing, but on larger-scale tasks like ImageNet it underperforms cross-entropy training, indicating that the benefits of the closed-form update may not overcome the loss function mismatch when the number of classes is very large or the problem is more complex. This limits the method’s applicability in scenarios where using the true cross-entropy loss (or others like hinge, etc.) is essential.

 - Theoretical Gaps for Finite Networks: The convergence guarantee provided (Theorem 4) is in the infinite-width NTK regime, which is a strong assumption that may not hold in practice. For finite networks, the loss $L^(\theta)$ can have non-global critical points, such as trivial feature representations that zero out gradients. The paper does not prove that gradient-based training will avoid these bad critical points in general finite settings. While standard deep networks typically don’t get stuck in completely uninformative representations, it’s theoretically possible that this new training objective could introduce different failure modes. The authors do not report observing such issues in practice (and indeed their method found good solutions in experiments), but a formal understanding for finite width is lacking. This leaves a slight gap in the theoretical guarantees while in contrast, conventional end-to-end training has well-understood critical point structures in overparametrized settings (e.g. no bad local minima under certain assumptions), it’s less clear for the $L^(\theta)$ objective.

 - Computational Overhead: Another concern is the extra computation required for the closed-form updates. Each iteration involves solving a d×d linear system (or inverting a matrix of size equal to the feature dimension) to compute $W^*(\theta)$. In the experiments, the feature dimension (d) was modest (e.g. 256 or 512), and this overhead was manageable, but for very high-dimensional features or extremely large output layers, this step could become a bottleneck. The paper does not report runtime comparisons or discuss strategies to mitigate this cost. It’s possible to use efficient linear algebra or warm-start techniques (given successive updates are on similar data), but such optimizations are not explored. Therefore, the scalability of the approach to very large networks or datasets (where d or the number of classes is in the thousands) remains a bit unclear.

 - Hyperparameter Sensitivity: The introduction of the proximal regularization coefficient $\lambda$ (and the interaction with the ridge parameter β, if any) adds an extra hyperparameter that needs tuning. The paper does include an ablation showing how performance varies with $\lambda$ and with the ridge coefficient. It appears that the method’s performance can be sensitive to the choice of $\lambda$, especially for smaller batch sizes where this term is critical – too large $\lambda$ might slow adaptation of the last layer, too small might reintroduce instability. The need to tune $\lambda$ (and potentially β and learning rate jointly) makes the method a bit more complex to use than standard SGD (which typically only needs a learning rate schedule and perhaps a weight decay). While this is not a major flaw, it means practitioners must budget some effort for hyperparameter search to fully realize the benefits of the approach.

 - Combining with Other Optimizers: The study found that using Adam (an adaptive optimizer) for the backbone parameters degraded performance relative to SGD. This suggests the closed-form last layer strategy might not be trivially compatible with all optimizer styles. The authors hypothesize that Adam’s internal state (momentum of gradients) might conflict with the idea of immediately resetting the last layer to optimal each time. Similarly, one might wonder if adding momentum to the backbone SGD could interfere with the two-timescale dynamics. The paper focuses on basic SGD; the limited exploration of optimizer variants is a minor weakness, as it leaves unclear whether the method can reap benefits from momentum or adaptive learning rates (which are often important in state-of-the-art training regimes). It may be that the closed-form update provides sufficient acceleration that such techniques are less important, but some discussion or experiments on this would strengthen the work.

**Questions:**

1. The closed-form last layer update requires solving a linear system at every iteration. Have the authors considered more efficient or scalable ways to implement this? For instance, could one exploit the approximate Kalman filter interpretation to update $W$ incrementally (reusing the previous inverse or using Sherman-Morrison updates) instead of recomputing from scratch each time? Any discussion on how the method scales with increasing feature dimension or number of classes would be valuable – e.g. is the matrix inversion step ever a bottleneck in practice, and how might one mitigate this for very large models?

2. The analysis shows that $L(\theta)$ can have spurious critical points (e.g. the feature map producing zero outputs is stationary). In practice, did the authors ever observe training getting “stuck” in a bad state, or does random initialization and SGD dynamics reliably avoid those trivial solutions? It would be helpful if the authors could elaborate on why, despite the non-convexity of $L(\theta)$, the method seemed to find good solutions (especially in finite-width networks not covered by NTK theory). Are there any conditions or initialization strategies needed to ensure convergence to a good optimum when using this closed-form approach?

3. The introduction of the proximal regularization coefficient $\lambda$ raises the question of how to choose it. The paper provides an ablation, but could the authors offer more guidance on this? For example, should $\lambda$ be scaled with the batch size or learning rate in some way? Is it essentially acting as an “effective batch size” or memory factor for the last layer updates? Additionally, the method still includes the ridge regularizer β in principle – in the experiments, was β set to zero (relying only on $\lambda$), or did the authors keep a small β as well? Clarifying the role of β versus $\lambda$ in practice (and whether one can simply set β=0 and treat $\lambda$ as a replacement) would help practitioners understand how to configure the training objective.

4. Since the closed-form strategy is tied to the squared loss, have the authors considered approximating or extending it to other losses? For instance, is there an analogue for cross-entropy (perhaps using a softmax pseudo-inverse or a one-step Newton update for the last layer)? The results on CIFAR-100 are intriguing in that the squared-loss with closed-form updates actually outperformed cross-entropy SGD. In contrast, on ImageNet the cross-entropy still had the edge. What do the authors believe explains this difference? Is it the larger number of classes, the maturity of cross-entropy tuning, or something about the loss landscapes? Any insight here could point to how one might combine the benefits of closed-form updates with the cross-entropy loss (or whether that is a promising direction at all). It would be interesting to know if a hybrid approach was attempted, for example, training with the closed-form $\ell_2$ method and then fine-tuning or calibrating with cross-entropy, and how that performed.

5. The finding that Adam performed worse than SGD in this framework is thought-provoking. Could the authors shed more light on why an adaptive optimizer or momentum might interfere with the closed-form last layer updates? Does the closed-form update essentially act like a large adaptive step for the last layer, making additional momentum unnecessary or even harmful for the backbone? It would be useful to know if the authors tried variants like adding momentum to the backbone SGD, or if they have recommendations on optimizer choice. Understanding this could help users avoid combinations that degrade performance, and it might reveal interesting interactions between fast two-timescale updates and optimizer internal dynamics.

---

> ### Author Response · Authors · 2025-11-14
> **Rebuttal part 1**
>
> We thank the reviewer JUgn for their feedback and for highlighting our main contributions. Please find our answer below.
>
> > Restriction to Squared Loss...
>
> We agree with this point. In future work we will focus on extending our methodology beyond the l2 loss.  We note, however, that the l2 loss is a well established loss in regression, and covers many interesting settings, including in causality and reinforcement learning.
>
> > Theoretical Gaps for Finite Networks...
>
> We agree that it would be interesting to provide further theoretical analyses in the finite width setting. For this, we envision an approach such as the one taken in Allen-Shu et al. A Convergence Theory for Deep Learning via Over-Parameterization (2019). This paper studies convergence for finite width, overparametrized setting. Our understanding is that it bridges the gap that the reviewer highlights, for classical convex losses. The NTK analysis that we provide, which focuses on our non-convex function space loss, is thus a first step toward such theory. We aim to investigate this further in future work, based on the insights collected in the present work.
>
>
> > Computational Overhead...
>
> We would like to highlight that the **complexity of our method is dominated by the feature dimension of the last layer (d)**. As you correctly pointed out, it is dominated by solving a linear d x d system in order to get in order to get (eq.13). This complexity comes from the fact that we need to use a batch covariance matrix, which is a matrix of size d x d. Importantly, **the complexity of our method depends only linearly on the output dimension o (i.e. on a number of classes)**. This is because we essentially share the covariance matrix across all the output dimensions. **Therefore, using our method in cases where we have a large number of classes is not prohibitive. The only potential computational bottleneck is the feature dimension (d).**
>
> Above, in the common response to all the reviewers, we have added the detailed metrics for wall clock times of our method in comparison to cross entropy on CIFAR-100.
>
> The main take-aways are:
>
> * As we increase the dimensionality (d) of the last layer, the speed of our method decreases in comparison to cross entropy.
>
> * It can decrease significantly (by 2x) only for small batch sizes. For large batch sizes, it only decreases by around 10-15% compared to cross entropy.
>
> * As the model size increases, the computation of the last layer becomes less expensive compared to the backbone, so the relative speed of our method increases.
>
> Please refer to the detailed table above. We will add the section about the computational cost in the appendix of our paper.
>
> In future work, we will consider mitigation strategies to increase the speed of our method. For example, instead of using full-covariance on the batch, we can use a low-rank approximation to it.
>
> > Hyperparameter Sensitivity...
>
> We agree that hyperparameter choice presents a potential challenge for our method. However in practice, based on our ablation experiment (Figure 6), we would argue that the method is not overly sensitive to choices of beta and lambda. We recommend the use of both a large lambda and beta (>= 10). Moreover we never use lambda and beta jointly: lambda is used in the proximal closed form loss, whereas beta is used in the ridge closed form loss. That is, we never add more than one additional hyperparameter. In our answer to the reviewer’s question 3 below, we give further insights into the choice of lambda, coming from the Kalman filter interpretation of the algorithm.
>
> > Combining with Other Optimizers...
>
> Regarding the momentum in the backbone SGD, in our CIFAR-100 and ImageNet experiments, we have already tried using momentum in the backbone. We found that adding / removing momentum did not lead to a qualitatively different behavior of our method, therefore momentum is not a problem. We believe that the main problem with Adam is the adaptive nature of the step size. We believe that in order to make our approach work effectively with Adam, an adaptive strategy of the parameter $\lambda$ must be proposed. This is what we envision for future work.

---

> ### Author Response · Authors · 2025-11-14
> **Rebuttal Part 2**
>
> > The closed-form last layer update requires solving a linear system at every iteration...could one exploit the approximate Kalman filter interpretation...
>
> Please see our response above regarding the complexity of the method wrt number of classes and last layer dimensionality.
>
> We have thought about using a Kalman filter interpretation. Such interpretation would require incrementally updating the precision/covariance matrix of the last layer distribution, and then use it to update the mean. In case when the Kalman filter is diagonal, this can be done efficiently using Sherman-Morrison updates, see [1] for example, in order to avoid matrix inversion. However, if we assume that the Kalman filter is not diagonal, we cannot avoid matrix inversion and computational complexity might become prohibitive. Moreover, using diagonal KF may restrict expressivity of the method since it will only consider diagonal empirical covariance. In future work, we plan to explore connections of our method to Kalman Filter and to its efficient approximations such as low-rank plus diagonal KF, see [2] for example.
>
>
> [1] Kalman Filter for Online Classification of Non-Stationary Data,  Michalis K. Titsias, Alexandre Galashov, Amal Rannen-Triki, Razvan Pascanu, Yee Whye Teh, Jorg Bornschein, 2023
>
> [2] Low-rank extended Kalman filtering for online learning of neural networks from streaming data Peter G. Chang, Gerardo Durán-Martín, Alexander Y Shestopaloff, Matt Jones, Kevin Murphy
>
> > The analysis shows that can have spurious critical points...Are there any conditions or initialization strategies needed to ensure convergence to a good optimum when using this closed-form approach?
>
> This is an interesting point. In practice we did not experience any such problems, and our intuition from the NTK result is that this loss is just as well-conditioned as the classical l^2 loss, but with fewer parameters to train (no last layer). Something we did observe with the proximal algorithm was sensitivity to the initialisation: a zero initialisation on the last layer is preferred, as otherwise the random initialization gets propagated to later solutions during training through the regularizer.
>
> >  The introduction of the proximal regularization coefficient raises the question of how to choose it...
>
> Thank you for these important remarks. When using the proximal loss with the hyperparameter $\lambda$, there is no need for a second parameter $\beta$. The hyperparameter $\beta$ is necessary when using the non-proximal closed form loss, as it stabilizes the training.
> Regarding a principled choice of $\lambda$, the Kalman filter interpretation of the loss in Appendix A hints at this. In this interpretation we see that $\lambda = \sigma_Y^2 / \sigma_W^2$, which is independent of the batch size, but depends on the learning rate. The greater the learning rate, the greater $\sigma_W^2$ (see equation 23), so the smaller $\lambda$. The intuition is that when the steps of gradient descent are bigger, we should move further from the previous last layer solution, and so regularize less. On the other hand $\lambda$ is independent of the batch size, because we see that when the batch size is bigger, the likelihood term in the loss already takes this into account (the sum will be bigger in equation 25), so the regularizer will be smaller in proportion without any change in $\lambda$. In Figure 6 we indeed see that choices of $\lambda >= 10$ work well regardless of the batch size. In the revised version we will explicitly state these insights that the Kalman filter/probabilistic interpretation provides.
>
> > Since the closed-form strategy is tied to the squared loss, have the authors considered approximating or extending it to other losses? For instance, is there an analogue for cross-entropy (perhaps using a softmax pseudo-inverse or a one-step Newton update for the last layer)?
>
> The naive application of Newton step will not scale well since it will require to compute the hessian of dimensionality (d*o x d*o), i.e., the product of feature and output dimension, squared. Our method, only uses the data covariance of shape (d * d) – i.e., it **shares the same covariance for all the output dimensions**. In future work, we will apply our methodology to cross entropy loss in such a way that we also share the covariance / hessian across all the output dimensions.

---

> > ### Author Response · Authors · 2025-11-14
> > **Rebuttal Part 3**
> >
> > > The results on CIFAR-100 are intriguing in that the squared-loss with closed-form updates actually outperformed cross-entropy SGD. In contrast, on ImageNet the cross-entropy still had the edge. What do the authors believe explains this difference? Is it the larger number of classes, the maturity of cross-entropy tuning, or something about the loss landscapes? Any insight here could point to how one might combine the benefits of closed-form updates with the cross-entropy loss (or whether that is a promising direction at all). It would be interesting to know if a hybrid approach was attempted, for example, training with the closed-form method and then fine-tuning or calibrating with cross-entropy, and how that performed.
> >
> > We emphasize that our method presently focuses on the l2 loss, which is not optimal for discrete likelihoods. Indeed, the models / training recipes on classification tasks have been optimized for cross entropy loss functions, so the “maturity” point is a likely factor. That said, we believe that the large number of classes could be specifically problematic for our method.
> >
> > We ran a simple experiment on CIFAR-100 where we track the maximum probability of a correct class. For l2 loss, we renormalize it to be positive (since the output of the model doesn't have to be positive nor normalized to 1), i.e., probabilities are given by $\tilde{p} = f(x) - \min_{c} f(x)_{c}$, $p=\frac{\tilde{p}} /sum_{c} \tilde{p}_c$, where $f(x)$ is the output of our model.
> >
> > The results are given in the following table:
> >
> > | Batch size | Cross entropy max probs | Ours max probs |
> > |:---:|:---:|:---:|
> > | 32 | 0.99 | 0.63 |
> > | 128 | 0.99 | 0.62 |
> > | 1024 | 0.99 | 0.73 |
> > | 4096 | 0.99 | 0.76 |
> >
> > We see that our method has much lower maximum probabilities compared to cross entropy, meaning that the results are less confident. This could be, in principle, problematic for datasets with large numbers of classes - we will report this result accordingly in the final version. In future work, we will investigate empirical strategies to improve this performance.
> >
> > > The finding that Adam performed worse than SGD in this framework is thought-provoking. Could the authors shed more light on why an adaptive optimizer or momentum might interfere with the closed-form last layer updates? Does the closed-form update essentially act like a large adaptive step for the last layer, making additional momentum unnecessary or even harmful for the backbone? It would be useful to know if the authors tried variants like adding momentum to the backbone SGD, or if they have recommendations on optimizer choice. Understanding this could help users avoid combinations that degrade performance, and it might reveal interesting interactions between fast two-timescale updates and optimizer internal dynamics.
> >
> > We did not find that momentum in the backbone interfered with our method. However, we did find that using Adam optimizer made the results worse. Our main hypothesis is that this is due to the adaptive nature of step size in Adam optimizer, while in our approach we keep $\lambda$ fixed. In future work, we will explore the adaptive strategies for $\lambda$ and we expect that this will help our method to work effectively with Adam.
> >
> > We hope the above addresses the reviewer concerns - if we have done so, we would be grateful if the reviewer might consider increasing their score.  We are happy to address any further questions.

---

### Official Review · Reviewer_k9oX · 2025-10-31

**Soundness:** 2
**Presentation:** 3
**Contribution:** 1
**Rating:** 2
**Confidence:** 4

**Summary:**

This paper exploits the fact that the square loss of a neural network is typically quadratic as a function of the last layer weights, therefore optimal values of the weights can be expressed in closed form (as a function of the other parameters).
They show that optimization of the other parameters does not require differentiating through the closed form solution, and provide a few fixes to the problem of overfitting small batches by the solution.
Some experiments on PDE and image classification show that the proposed method outperforms SGD on square loss, but it does not outperform standard cross entropy in most cases.

**Strengths:**

- The paper is extremely clear and straightforward.

- The method is very easy to implement.

**Weaknesses:**

- The idea is not quite new, other papers had used it even if some details are different.

- Theoretical contributions are weak. Theorem 1 and 2 are trivial. I’m sure you can find them in many previous papers, perhaps in slightly different contexts. They are not really “Theorems”, they are just “chain rule.”

- The section about NTK and Theorem 3 and 4 are also trivial. It’s obvious that a non-convex loss may have stationary points that are not global minimisers, and that is resolved by the NTK. This is all very well known.

- Experimental results are underwhelming. Although the comparison with SGD on square loss is somewhat fair from the scientific point of view, it’s very weak, and it's not clear whether this method will be ever be practical at all. Cross-entropy wins most of the time, and it’s not clear what would happen when the method is compared with and/or extended to other optimisers that outperform SGD by large amounts.

Minor:

- I disagree with the statement: “Without the regularization to previous last layer solutions, our method is analogous to putting a large learning rate on the last layer.” If you put a large learning rate on W then optimization of W destabilize, even if loss is quadratic. Instead, your method is equivalent to a Newton step, that converges to the minimum of a quadratic loss in one step.

- I believe Eq.(21) has a typo in the LHS second term, it should be a first derivative.

- In the non-stochastic setting, it should be possible to show that Eq.(9) is necessarily better than standard gradient descent. That would have been a nice result (although still quite unsurprising).

**Questions:**

NA

---

> ### Author Response · Authors · 2025-11-14
> **Rebuttal**
>
> We thank the Reviewer k9oX for their feedback. Please see our detailed answer below.
>
> > The idea is not new, other papers had used it...
>
> While the idea of using a closed-form solution in the last layer is known in the theoretical community (which we cite [1,2,3,4]), we are the first to put it into a practical algorithm. A naive approach of using it would be to not use a proximal penalty and simply use a ridge regression, eq.10. One of the crucial contributions of our work is noticing that closed form proximal updates on the last layer is similar to SGD steps, thus motivating the use of a proximal penalty term. Please also see our literature review for more details. If there exists specific prior work to which you would like to draw our attention, we would be grateful if you could provide us with the associated references.
>
> [1] Pierre Marion and Raphaël Berthier. Leveraging the two-timescale regime to demonstrate convergence of neural networks. (2023)
>
> [2] Raphaël Berthier, Andrea Montanari, and Kangjie Zhou. Learning Time-Scales in Two-Layers Neural Networks. (2024)
>
> [3] Alberto Bietti, Joan Bruna, and Loucas Pillaud-Vivien. On learning Gaussian multi-index models with gradient flow part I: General properties and two-timescale learning. (2025)
>
> [4] Raphaël Barboni, Gabriel Peyré, and François-Xavier Vialard. Ultra-fast feature learning for the training of two-layer neural networks in the two-timescale regime. (2025)
>
>
> > Theoretical contributions are weak...
>
> The reviewer is correct in saying Theorem 1 is not novel – we included a citation. See also the related works section, where we cited several more papers of interest. While Theorems 1 and 2 are mathematically straightforward, their practical implications are significant. A naive implementation of the idea would require doing backpropagation through the solution of linear regression into backbone parameters. Such backpropagation results in a much costlier and more numerically unstable method. This naive implementation was used for example in the original DFIV paper (Xu et al. (2020)). Theorem 1 shows that we can skip backpropagation in the full-batch setting and Theorem 2 shows that we can do it in the mini-batch one. Essentially, these theorems justify the use of coordinate descent strategy used, which is key to designing a practical algorithm.
>
> > The section about NTK and Theorem 3 and 4 are also trivial...
>
> While it is known that non-convex loss may have stationary points that are not global minimisers, we are not aware of any statements in the literature that directly relate to Theorems 3 and 4. Specifically, Theorem 3 is about studying the loss of interest. We show it is non-convex and characterize its critical points, providing insights into its landscape (see Appendix C). With this characterization, we are able to show that the non-minimizing critical points are almost surely not attained in the NTK regime, and convergence to a global minimum is thus guaranteed. The proof uses the specific form of the critical points (see Appendix C1), as classical convergence results in the NTK regime guarantees convergence to a critical point, not a global minimum for general losses. The intuition for why it works in our case is that the non-minimizing critical points of this loss correspond to a low rank manifold which is avoided when the NGPK is positive definite. This is unlike the usual results for squared and cross entropy losses, where (function space) convexity can be invoked to directly prove convergence to a global minimum. Again, we would be happy to review any additional specific prior references that you can recommend on this topic.
>
> > Exp. results are underwhelming...
>
> In our experiments, we provide comparisons to SGD with squared loss (regression and instrumental variable regression), where we consistently see that our method outperforms SGD. Moreover, in case of cross entropy experiments, we observe that our method (which relies on l2 loss) is competitive to cross-entropy and outperforms the naive use of the l2 loss. The point of these experiments is to first highlight that our method can outperform the naive use of the l2 loss, but also could in principle be used outside of the l2 loss assumption. Our recommendation for practitioners is to use this method when they are trying to solve a regression problem. Extending the methodology to categorial likelihoods will be the focus of our future work.
>
> > ...method is equivalent to a Newton step, that converges to the minimum of a quadratic loss in one step....
>
> Our original sentence was indeed imprecise, we thank you for pointing it out and providing a better alternative. We have made the change in the revised version.
>
> > Typo in eq.21
>
> Thank you for pointing this out, we have now fixed this.
>
> > ...should be possible to show that Eq.(9) is necessarily better than standard gradient descent...
>
> This is an interesting point, we thank the reviewer for raising it and aim to investigate it in future work.

---

### Official Review · Reviewer_A5q7 · 2025-11-07

**Soundness:** 3
**Presentation:** 2
**Contribution:** 2
**Rating:** 6
**Confidence:** 3

**Summary:**

The paper introduces new optimization methods that treat the last layer weights as an explicit function of the backbone parameters and optimize this function with respect to only the backbone parameters. The authors also extend this framework to a stochastic gradient descent (mini-batch) version. They prove the convergence guarantee under the Neural Tangent Kernel (NTK) regime. Experimental results demonstrate that the proposed methods perform better compared to baseline SGD across a variety of tasks.

**Strengths:**

- Rigorous derivation of closed-form last-layer optimization methods.
- The extension to the SGD mini-batch setting with proximal loss is an interesting derivation.
- Empirical results demonstrate better efficiency and accuracy compared to standard SGD across multiple tasks.
- Theoretical analysis under the Neural Tangent Kernel (NTK) regime provides solid convergence guarantees in the infinite-width limit.

**Weaknesses:**

- Typos and formatting issues detract slightly from readability and presentation quality.
- The paper lacks a clear explanation of how the NTK regime theoretical analysis directly supports convergence guarantees of the two optimization methods introduced above.

Typos:
- In line 267, equation (18), the author should include the sum of all square losses
- Line 289-290: “the initial function neural network function $\phi$...”
- Line 293-294, “The following result shows that if we make the slightly stronger assumption $\textbf{that}$ the NGPK is positive definite...”

Figures and references format:
- Page 12 and page 21

**Questions:**

- In Figure 1, why does the curve for $W^\ast(\theta)$  appear to be multi-valued for some values of $\theta$, while later it is treated as a function of $\theta$? Could the authors clarify this discrepancy?
- In equation (13), should the matrices $X$ and $Y$ correspond to mini-batch subsets $\mathcal{B}_t$ instead of the full datasets?
- In the experiments, is the "$l_2$ c.f. ridge ($\beta$)" method implemented with mini-batches, or is it full-batch as suggested by the equation (9)?
- How does the NTK regime theoretical analysis relate to the convergence theory of the two optimization methods introduced before?

---

> ### Author Response · Authors · 2025-11-14
> **Rebuttal**
>
> We thank Reviewer A5q7 for their feedback and for pointing out typos in our manuscript. Please find below our response addressing your concerns.
>
> > Typos and formatting issues detract slightly from readability and presentation quality. Typos:
> In line 267, equation (18), the author should include the sum of all square losses
> Line 289-290: “the initial function neural network function ...”
> Line 293-294, “The following result shows that if we make the slightly stronger assumption  the NGPK is positive definite...”
>
> We thank the reviewer for noticing these typos. We fix them in the revised version of our paper.
>
> > In Figure 1, why does the curve for appear to be multi-valued for some values of , while later it is treated as a function of ? Could the authors clarify this discrepancy?
>
> Thank you for spotting this. We accidentally swapped the labels for W and theta, meaning \theta should be on the y-axis, and W on the x-axis. Then there is only one last layer W(\theta) per \theta. We have corrected this in the revised version.
>
> > In equation (13), should the matrices  and  correspond to mini-batch subsets  instead of the full datasets?
>
> You are correct, these should be mini-batches, not full data as implied by the notation. We will fix this discrepancy.
>
> > In the experiments, is the " c.f. ridge ( )" method implemented with mini-batches, or is it full-batch as suggested by the equation (9)?
>
> Thank you for noticing this. We indeed used a mini batch version, eq.10-eq.11. We will correctly refer to eq.10 and eq.11, and we will add an update rule of the form eq.9 to the stochastic setting after eq.11.
>
> > How does the NTK regime theoretical analysis relate to the convergence theory of the two optimization methods introduced before?
>
> The NTK analysis from section 5 provides convergence guarantees of the optimization method from equation 9, under essentially three simplifying assumptions:
>
> 1. Gradient flow assumption: we approximate the discrete step gradient descent with a continuous gradient flow,
>
> 2. Non-stochasticity: full batch setting
>
> 3. Infinite width regime: we consider the setting where the neural network has infinite width, in the NTK limit.
>
> Showing convergence in this regime is in our opinion a necessary step towards providing further convergence results that relax the assumptions above, and directly relate to practice. For standard losses, extensions to the finite width setting and stochastic (mini-batch) settings can be found for instance in [1,2]. Providing this analysis for the present loss is an interesting avenue of future research, which could provide guarantees for the proximal algorithm.
>
> We hope the above addresses the reviewer’s concerns - if we have done so, we would be grateful if the reviewer might consider increasing their score.  We are happy to address any further questions.
>
> **References**:
>
> [1] Rajat Dwaraknath, Tolga Ergen, Mert Pilanci. Fixing the NTK: From Neural Network Linearizations to Exact Convex Programs (2023)
>
> [2] Zeyuan Allen-Zhu, Yuanzhi Li, Zhao Song. A Convergence Theory for Deep Learning via Over-Parameterization (2019)

---

### Official Review · Reviewer_aGYw · 2025-11-10

**Soundness:** 3
**Presentation:** 3
**Contribution:** 3
**Rating:** 8
**Confidence:** 2

**Summary:**

This paper proposes an optimization method that leverages the closed-form solution for the linear last layer under a squared loss. The method treats the last layer as a function of the backbone parameters and optimizes only the backbone, which is shown to be equivalent to alternating gradient steps on the backbone with closed-form updates on the last layer.

**Problem formulation**

The paper considers a model $f(x;W,\theta)=W\phi_{\theta}(x)$, where $\phi_{\theta}$ is the neural network backbone and $W$ is the last linear layer. Since the optimal $W^{\star}(\theta)$ for a fixed $\theta$ is known in closed-form via ridge regression, the authors reformulate the problem to optimize the backbone parameters $\theta$ by minimizing the loss $\mathcal{L}^{star}(\theta):=\mathcal{L}(W^{\star}(\theta),\theta)$.

**Main results. Provide one or two sentence summary**

The key theoretical result is that optimizing this reformulated loss does not require backpropagation through the complex closed-form solution; by the envelope theorem, the gradient $\nabla_{\theta}\mathcal{L}^{*}(\theta)$ is simply $\nabla_{\theta}\mathcal{L}(W^{\star},\theta)$. This property is extended to a practical stochastic (proximal) version of the loss, and the method is proven to converge to a global minimum in the NTK regime.

**Technical approach**

To adapt the method for stochastic gradient descent and prevent overfitting the last layer to minibatches, the authors introduce a proximal loss that regularizes the batch solution against the previous last layer estimate $W_t$. The practical algorithm (Algorithm 1) first updates the backbone parameters $\theta_t$ via a standard gradient step (using $W_{t-1}$), and then computes the new last layer $W_t$ using the closed-form solution of this proximal loss based on the current batch and updated backbone $\theta_t$.

**Experiment**

The proposed proximal method is shown to outperform standard SGD on a squared loss across regression (Fourier Neural Operators, DFIV) and classification (CIFAR, ImageNet) tasks. The approach is particularly effective and stable across all batch sizes, unlike a naive closed-form ridge solution which performs poorly on small batches.

**Strengths:**

The paper develops a practical and stable proximal-based algorithm that effectively leverages the closed-form last layer solution to accelerate training in stochastic, small-batch settings where naive closed-form updates would otherwise fail. Both theory and experiments are solid

**Weaknesses:**

.

**Questions:**

.

**Details Of Ethics Concerns:**

.

---

> ### Author Response · Authors · 2025-11-14
> **Rebuttal**
>
> We thank Reviewer aGYw for their positive feedback on our paper.
>
> The reviewer highlighted our theoretical contributions, that optimizing the reformulated loss does not require backpropagation through the complex closed-form solution, as well as our result about global convergence in the NTK regime. Moreover, the reviewer noted our contribution of using a proximal loss for the stochastic setting, which consists in regularizing for previous last layer solutions. As they noticed, our approach outperforms standard SGD on a squared loss across regression and classification tasks. Our approach is particularly effective and stable across all batch sizes, unlike a naive closed-form ridge solution which performs poorly on small batches. We remain available for any further questions.

---

### Author Response · Authors · 2025-11-14
**Common rebuttal comment.**

We thank all the reviewers for their feedback. Below, we provide wall-clock times for our method on CIFAR-100. We will add these tables to the revised version of the paper.

# Wall-clock times on CIFAR-100

We consider ResNet18 and ResNet50 backbones, where add an additional last layer of a given dimensionality (**Last Layer Dim**). We report steps per second (SPS) metrics as well as total time (Time) for training a model for 100 epochs on CIFAR-100 using either our method (Algorithm 1) or cross entropy (CE). We report metrics for different batch sizes and last layer dimensions. Moreover, we report `rSPS = SPS(OURS) / SPS(CE)` (higher means our method is faster than CE) and `rTime = Time(Ours) / Time (CE)` (lower means our method is faster than CE). We use A100 GPU.

**Take-aways**:

First, we observe that for small batch size (32), rTime of our method increases from 0.92 (last layer dim = 128) to 2.37 (last layer dim = 4096) for ResNet18, and from 1.66 (last layer dim = 128) to 1.84 (last layer dim = 4096) for ResNet50. This means that for small batch sizes, as we increase the last layer dimension, our method becomes significantly slower than cross entropy. However, we also notice that the rTime decreases as we increase the model size from ResNet18 to ResNet50. This highlights the fact that as the model size increases, the computation required for the last layer becomes relatively smaller compared to the computation of the backbone.

Second, we observe that for large batch size (1024), `rTime` only increases from 1.09 (last layer dim = 128) to 1.18 (last layer dim = 4096) for ResNet18; and from 1.16(last layer dim = 128) to 1.20 (last layer dim = 4096). Finally, for ResNet18, we see that for batch size = 4096, `rTime` basically stays very similar (from 1.12 to 1.14). This highlights that in the large batch size regime, our method is roughly 10-15% slower than cross entropy.


## Detailed metrics for ResNet18

| Batch Size | Last Layer Dim | SPS (Ours) | SPS (CE) | rSPS (Ours) / (CE) | Time (Ours) | Time (CE) | rTime (Ours) / (CE) |
|:---:|:---:|:---:|:---:|:---:|:---:|:---:|:---:|
| 32 | 128 | 89.98 | 81.02 | 1.11 | 2164.20 | 2348.11 | 0.92 |
| 32 | 256 | 87.35 | 78.97 | 1.11 | 2214.96 | 2400.73 | 0.92 |
| 32 | 512 | 78.48 | 79.86 | 0.98 | 2413.57 | 2374.19 | 1.02 |
| 32 | 1024 | 70.23 | 81.89 | 0.86 | 2633.73 | 2325.33 | 1.13 |
| 32 | 2048 | 51.47 | 79.66 | 0.65 | 3426.96 | 2383.31 | 1.44 |
| 32 | 4096 | 30.41 | 82.44 | 0.37 | 5488.86 | 2318.55 | 2.37 |
| 128 | 128 | 40.57 | 41.24 | 0.98 | 1293.17 | 1277.97 | 1.01 |
| 128 | 256 | 39.59 | 40.74 | 0.97 | 1320.46 | 1281.52 | 1.03 |
| 128 | 512 | 38.77 | 40.72 | 0.95 | 1333.75 | 1282.12 | 1.04 |
| 128 | 1024 | 35.47 | 40.44 | 0.88 | 1422.86 | 1291.61 | 1.10 |
| 128 | 2048 | 29.76 | 41.14 | 0.72 | 1614.35 | 1278.58 | 1.26 |
| 128 | 4096 | 20.92 | 40.54 | 0.52 | 2140.42 | 1294.91 | 1.65 |
| 1024 | 128 | 4.66 | 5.07 | 0.92 | 1055.30 | 967.64 | 1.09 |
| 1024 | 256 | 4.68 | 5.04 | 0.93 | 1046.71 | 975.81 | 1.07 |
| 1024 | 512 | 4.68 | 5.09 | 0.92 | 1050.68 | 966.34 | 1.09 |
| 1024 | 1024 | 4.66 | 5.05 | 0.92 | 1051.08 | 972.14 | 1.08 |
| 1024 | 2048 | 4.57 | 5.10 | 0.89 | 1072.17 | 963.36 | 1.11 |
| 1024 | 4096 | 4.21 | 5.02 | 0.84 | 1164.10 | 982.72 | 1.18 |
| 4096 | 128 | 1.11 | 1.24 | 0.89 | 1084.92 | 966.99 | 1.12 |
| 4096 | 256 | 1.09 | 1.24 | 0.88 | 1097.92 | 968.55 | 1.13 |
| 4096 | 512 | 1.08 | 1.24 | 0.87 | 1105.35 | 963.45 | 1.15 |
| 4096 | 1024 | 1.10 | 1.23 | 0.90 | 1089.28 | 973.90 | 1.12 |
| 4096 | 2048 | 1.09 | 1.23 | 0.89 | 1100.39 | 974.79 | 1.13 |
| 4096 | 4096 | 1.08 | 1.24 | 0.88 | 1107.10 | 971.15 | 1.14 |


## Detailed metrics for ResNet50

| Batch Size | Last Layer Dim | SPS (Ours) | SPS (CE) | rSPS (Ours) / (CE) | Time (Ours) | Time (CE) | rTime (Ours) / (CE) |
|:---:|:---:|:---:|:---:|:---:|:---:|:---:|:---:|
| 32 | 128 | 36.96 | 42.94 | 0.86 | 4735.09 | 2855.43 | 1.66 |
| 32 | 256 | 41.18 | 42.84 | 0.96 | 4240.65 | 4056.31 | 1.05 |
| 32 | 512 | 35.85 | 39.59 | 0.91 | 4903.87 | 3496.42 | 1.40 |
| 32 | 1024 | 35.22 | 43.28 | 0.81 | 4903.89 | 4002.80 | 1.23 |
| 32 | 2048 | 30.85 | 41.52 | 0.74 | 5482.36 | 4173.49 | 1.31 |
| 32 | 4096 | 21.68 | 41.97 | 0.52 | 7605.07 | 4133.77 | 1.84 |
| 128 | 128 | 16.53 | 18.62 | 0.89 | 2645.24 | 2384.37 | 1.11 |
| 128 | 256 | 16.37 | 18.49 | 0.89 | 2659.99 | 2397.10 | 1.11 |
| 128 | 512 | 16.15 | 18.30 | 0.88 | 2698.11 | 2423.22 | 1.11 |
| 128 | 1024 | 15.45 | 18.52 | 0.83 | 2801.06 | 2405.96 | 1.16 |
| 128 | 2048 | 14.31 | 18.44 | 0.78 | 3000.61 | 2409.51 | 1.25 |
| 128 | 4096 | 11.95 | 18.05 | 0.66 | 3532.58 | 2448.23 | 1.44 |
| 1024 | 128 | 2.26 | 2.59 | 0.87 | 2154.44 | 1875.26 | 1.15 |
| 1024 | 256 | 2.25 | 2.61 | 0.86 | 2151.89 | 1866.88 | 1.15 |
| 1024 | 512 | 2.24 | 2.60 | 0.86 | 2163.26 | 1874.76 | 1.15 |
| 1024 | 1024 | 2.23 | 2.59 | 0.86 | 2174.04 | 1876.25 | 1.16 |
| 1024 | 2048 | 2.19 | 2.57 | 0.85 | 2216.16 | 1889.88 | 1.17 |
| 1024 | 4096 | 2.12 | 2.56 | 0.83 | 2286.04 | 1898.44 | 1.20 |

---

### Meta-Review · Area_Chair_iycW · 2025-12-24

**Summary:**

The reviewers had the following concerns:
1. Limited practical significance. The proposed method is using l2 loss, which can not compete with training with CE loss for the most of the experiments. One reviewer also points out several technical comments regarding the computation overhead, sensitivity to the regularization coefficient, and the compatibility with optimizers other than SGD.
2. Weak theoretical contributions. The theorems are either straightforward computation by chain rule, or direct application of existing analysis like NTK.

While I agree with the above concerns from the reviewers, I also found the quality of the review subpar. Based on my own reading of the paper, the major weakness in the paper is the empirical result, which fails to show the practical impact of the proposed algorithm. Specifically, the primarily goal of this paper is to propose a practical algorithm for training neural networks. Since the loss used is l2 in the training problem, so reasonable paths to show the practical usefulness of the algorithm include: (1) Conduct extensive experiments on regression problems in several task domains to show the efficiency/performance improvement; (2) Use l2 for classification problems anyway and show it outperforms CE in some cases. The paper did not conduct sufficient experiments in either paths, nor the rebuttal address this issue.

**Reviewer Concerns:**

I think both concerns are still outstanding after rebuttal.

**Reviewer Scores:**

I do not think the reviewer would change their score.

---

### Decision · Program_Chairs · 2026-01-26

Reject